# Invariant Causal Representation Learning for Generalization in Imitation and Reinforcement Learning

Chaochao Lu[1,2], Jośe Miguel Hernández-Lobato[1,3], Bernhard Schölkopf [2]

## Abstract

A fundamental challenge in imitation and reinforcement learning is to learn policies, representations, or dynamics that do not build on spurious correlations and generalize beyond the specific environments that they were trained on. We investigate these generalization problems from a unified view. For this, we propose a general framework to tackle them with theoretical guarantees on both identifiability and generalizability under mild assumptions on environmental changes. By leveraging a diverse set of training environments, we construct a data representation that ignores any spurious features and consistently predicts target variables well across environments. Following this approach, we build invariant predictors in terms of policy, representations, and dynamics. We theoretically show that the resulting policies, representations, and dynamics are able to generalize to unseen environments. Extensive experiments on both synthetic and real-world datasets show that our methods attain improved generalization over a variety of baselines.

## 1 Introduction

Imitation learning (IL) (Argall et al., 2009) concerns learning a policy via expert demonstrations without access to a reward function, while reinforcement learning (RL) (Sutton & Barto, 2018) focuses on learning a policy via interaction and reward feedback. One of the fundamental challenges with state-of-the-art IL and RL approaches is their limited ability to generalize outside of the specific environments they were trained on, because they learn easy-to-fit spurious correlations which are prone to change between training and testing environments.

A natural approach to generalization across environmental changes is constructing a representation that consistently predicts the target well across environments. That is, such a representation comprises only those features that describe true correlations of interest with the target that are stable across environments, excluding any features presenting environment-varying spurious correlations with the target. By exploiting the varying degrees of spurious correlations naturally present in training data collected from multiple environments, one can try to identify stable features and build invariant predictors (e.g., invariant policies and dynamics) that are able to generalize to unseen environments (Peters et al., 2016; Arjovsky et al., 2019; Lu et al., 2022). In both IL and RL, much effort has been made in this direction. Specifically, in the IL setting, prior work has shown that only a policy relying solely on the true causes of expert actions can robustly model the mapping from states to optimal/expert actions, which is stable under environmental shift (de Haan et al., 2019; Zhang et al., 2020c; Bica et al., 2021; Samsami et al., 2021; Kumor et al., 2021). In the RL setting, one usually considers three types of generalization problems. First, it has been shown that a representation consisting only of all the causal ancestors of the reward is the minimal sufficient representation for policy learning (Zhang et al., 2020a; Huang et al., 2022; 2021). This is because in RL, we seek to model *return* (i.e., cumulative reward) rather than solely rewards, which requires a representation that can capture multi-timestep interactions. Such a representation can generalize to unseen testing environments with unseen reward functions, as long as the new reward functions are causally dependent on a subset of the same causal ancestors that determine the original reward function in training environments (Zhang et al., 2020b; Huang et al., 2022). Second, in some scenarios where testing and training environments further share the same reward function, the policy relying solely on such representation, which is learned from training environments, can even directly generalize to unseen testing environments (Higgins et al., 2017; Harrison et al., 2020; Zhang et al., 2020b). Third, when the rewards between training and testing environments neither share the same function

---

[1]University of Cambridge, [2]Max Planck Institute for Intelligent Systems, [3]The Alan Turing Institute, Correspondence at cl641@cam.ac.uk.

nor depend on a subset of the same causal ancestors, it is generally impossible to generalize either representations or policies. However, it is still possible to learn generalizable (local) dynamics models by building invariant predictors on a per-state-variable level[1] (Tomar et al., 2021). Apparently, this also relies on identifying the true causes of each state variable from those state variables in the previous timestep. In this paper, to distinguish the three types of RL generalization problems above, we call them *representation generalization*, *policy generalization*, and *dynamics generalization*, respectively. A graphical illustration is presented in Fig. 3 of the appendix.

In this work, we attempt to look at the different generalization problems in IL and RL from a unified view, and propose a general framework, under more relaxed assumptions over the distributional shift and the underlying causal structures, to tackle them, with theoretical guarantees on both identifiability and generalizability. The key idea is that by taking advantage of structural relationships between environmental variables (i.e., observations, states, actions, and rewards), we formulate the generalization problems in both IL and RL in the framework of invariant causal representation learning (iCaRL) (Lu et al., 2022). This framework offers a tool, with theoretical guarantees under rather general assumptions over the underlying causal diagram, to first identify direct causes of a given target from data and then use those causes to build invariant predictors that are able to generalize to unseen testing environments, which will be briefly reviewed in Appendix C.1. While our methodology builds significantly on the supervised learning iCaRL framework, the application of this methodology to problems in RL and IL is new and the results obtained are promising.

Our contributions are summarised as follows: (1) We investigate different generalization problems in IL and RL in a unified perspective; (2) We propose a general framework to tackle the generalization problems in IL and RL, with theoretical guarantees on both identifiability and generalizability; (3) We propose general assumptions over the distributional shift and the underlying causal structures, covering many real-world scenarios in IL and RL; (4) We show that our framework has theoretical guarantees for OOD generalization in terms of policy, representation, and dynamics in IL and RL.

## 2 GENERALIZATION IN IMITATION LEARNING

### 2.1 PROBLEM FORMULATION

We consider the BC approach (Widrow, 1964; Pomerleau, 1989; 1991; Bain & Sammut, 1995; Schaal, 1999; Muller et al., 2006; Mülling et al., 2013; Bojarski et al., 2016; Mahler & Goldberg, 2017; Bansal et al., 2018) to learning an imitation policy on the basis of expert demonstrations collected from multiple environments, with the aim to generalize it to unseen environments. Technically, we consider a family of environments $\mathcal{M}_{\mathcal{E}_{all}} = \{(\mathcal{X}^e, \mathcal{A}, P^e, R^e, \gamma) \, | \, e \in \mathcal{E}_{all}\}$, with observations $\mathbf{x}^e \in \mathcal{X}^e \subseteq \mathbb{R}^d$, actions $\mathbf{a} \in \mathcal{A}$, a transition dynamics $P^e \equiv p^e((\mathbf{x}')^e | \mathbf{x}^e, \mathbf{a})$, a reward function $R^e(\mathbf{x}^e, \mathbf{a}) \in \mathbb{R}$, and a discount factor $\gamma \in [0, 1)$. Note that, the action space and discount factor do not change across all environments $\mathcal{E}_{all}$. We assume only access to a dataset of recorded demonstrations $\mathcal{D}_{\mathcal{E}_{tr}} = \{\{\tau_i^e\}_{i=1}^{N_e} \, | \, e \in \mathcal{E}_{tr}\}$ from a set of training environments $\mathcal{E}_{tr} \subset \mathcal{E}_{all}$, where each demonstration $\tau^e$ consists of a sequence of environment specific observation-action pairs $\tau^e = (\mathbf{x}_t^e, \mathbf{a}_t)_{t=0,...}$ that are drawn from an expert policy $\pi_*$. Our goal is to learn a policy $\pi$ from $\mathcal{D}_{\mathcal{E}_{tr}}$ so that it is able to mimic the expert behaviour across $\mathcal{E}_{all}$ that share a certain structure. Specifically, we seek a policy that generalizes well across $\mathcal{E}_{all}$ by solving the optimization problem:

$$\min_{\pi} \max_{e \in \mathcal{E}_{tr}} \sum_{\tau^e \in \mathcal{D}_e} \sum_{(\mathbf{x}_t^e, \mathbf{a}_t) \in \tau^e} \ell^e(\pi(\mathbf{x}_t^e), \mathbf{a}_t), \tag{1}$$

where each $\ell^e$ is a choice of environment specific loss function and $\mathcal{D}_e = \{\tau_i^e\}_{i=1}^{N_e}$ is a set of expert demonstrations collected from the training environment $e \in \mathcal{E}_{tr}$. Unless stated otherwise, for simplicity of notation, we drop the superscript $e$ when referring to the union over all the environments $\mathcal{E}_{all}$ in the rest of this paper.

As discussed in Section 1, prior work has shown that only a policy replying solely on the true causes of expert actions can generalize well to $\mathcal{E}_{all}$. Hence, solving (1) is reduced to discovering the causal features for expert actions from observations across $\mathcal{E}_{tr}$, and then, using them, build an invariant policy. In the IL context, this can be also interpreted in the way that a generalizable policy depends only on some shared components of the true latent states denoted by $\mathbf{s}_t$, rather than their corresponding observations $\mathbf{x}_t$, of the environments (cf. Fig. 3). Note that, in this work we do not address the partial-observability problem explicitly (Thrun et al., 2005). Instead, we approximate

---

[1]This is because in many real world scenarios, each state variable only depends on a small subset of those state variables in the previous timestep, and because spurious correlations arise for individual state variable dynamics.

stacked consecutive observations as the observation $\mathbf{x}_t$, that is, $\mathbf{x}_t$ loses no information about $\mathbf{s}_t$ (Hausknecht & Stone, 2015; de Haan et al., 2019; Gelada et al., 2019; Zhang et al., 2020b):

**Assumption 1** (Full-observability). *Observations* $\mathbf{x}$ *contain all the information about states* $\mathbf{s}$.

### 2.1.1 ASSUMPTIONS ON THE CAUSAL DIAGRAM

We denote the data representation (true state) of observation $\mathbf{x}_t$ by $\mathbf{s}_t = (s_t^{p_1}, \ldots, s_t^{p_r}, s_t^{c_1}, \ldots, s_t^{c_k}) \in \mathbb{R}^n$, where $n = r + k$, and $\{s_t^i\}_{i \in I_p \doteq \{p_1, \ldots, p_r\}}$ and $\{s_t^j\}_{j \in I_c \doteq \{c_1, \ldots, c_k\}}$ are multiple scalar *causal factors* and *non-causal factors* of $\mathbf{a}_t$, respectively. We denote $\mathbf{s}_t^p \doteq (s_t^{p_1}, \ldots, s_t^{p_r})$ and $\mathbf{s}_t^c \doteq (s_t^{c_1}, \ldots, s_t^{c_k})$ for the ease of clarification. It is assumed that $\mathbf{s}$ is of lower dimension than $\mathbf{x}$, that is, $n \leq d$. We assume that all the environments $\mathcal{E}_{all}$ share some latent structure between $\mathbf{x}_t$, $\mathbf{s}_t$ and $\mathbf{a}_t$, and consider different degrees to which this structure may be shared, which is encapsulated in Fig. 1 and summarized as below. Note that, at this moment we do not explicitly consider the node $\mathbf{c}_{t-1} \doteq (\mathbf{s}_{t-1}, \mathbf{a}_{t-1})$ and its both incoming and outgoing arrows (marked in blue), and defer them to Appendix F.1.

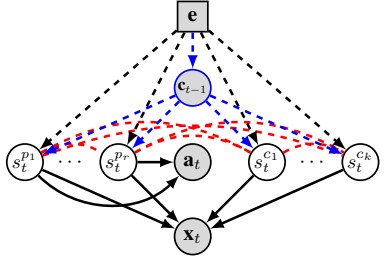

Figure 1: Causal structure in the IL settings, where we assume there exist multiple unobserved state variables. Each of them could be either a parent of $\mathbf{a}_t$, or has no direct connection with $\mathbf{a}_t$. We allow for arbitrary connections between the latent state variables (red dashed lines) as long as the resulting causal diagram including $\mathbf{a}_t$ is a DAG.

**Assumption 2.** *(a) The causal graph containing* $\mathbf{s}_t$ *and* $\mathbf{a}_t$ *is a DAG; (b)* $\mathbf{x}_t \perp\!\!\!\perp \mathbf{a}_t, \mathbf{e} | \mathbf{s}_t$, *implying that* $p(\mathbf{x}_t | \mathbf{s}_t)$ *is invariant across* $\mathcal{E}_{all}$; *(c)* $\mathbf{a}_t \perp\!\!\!\perp \mathbf{e} | \mathbf{s}_t^p$, *implying that* $p(\mathbf{a}_t | \mathbf{s}_t^p)$ *is invariant across* $\mathcal{E}_{all}$.

Assumption 2, together with its corresponding causal diagram in Fig. 1, is flexible enough to cover most scenarios in the IL setting, which is detailedly explained in Appendix F.2.

### 2.1.2 ASSUMPTIONS ON THE PRIOR

When the underlying causal diagram across $\mathcal{E}_{all}$ satisfies Assumption 2b, it is straightforward to obtain the following primary assumption over the prior $p(\mathbf{s}_t | \mathbf{a}_t, \mathbf{e})$ leading to identifiability of the latent variables $\mathbf{s}_t$ by directly substituting $(\mathbf{z}, \mathbf{y})$ with $(\mathbf{s}_t, \mathbf{a}_t)$ in Assumption 2 of Lu et al. (2022) (cf. Assumption 10 in Appendix C.1).

**Assumption 3.** $p_{\boldsymbol{T}, \boldsymbol{\lambda}}(\mathbf{s}_t | \mathbf{a}_t, \mathbf{e})$ *belongs to a general exponential family with parameter vector given by an arbitrary function* $\boldsymbol{\lambda}(\mathbf{a}_t, \mathbf{e})$ *and sufficient statistics* $\boldsymbol{T}(\mathbf{s}_t) = [\boldsymbol{T}_f(\mathbf{s}_t)^{\mathrm{T}}, \boldsymbol{T}_{NN}(\mathbf{s}_t)^{\mathrm{T}}]^{\mathrm{T}}$ *given by the concatenation of a) the sufficient statistics* $\boldsymbol{T}_f(\mathbf{s}_t) = [\boldsymbol{T}_1(s_t^1)^{\mathrm{T}}, \cdots, \boldsymbol{T}_n(s_t^n)^{\mathrm{T}}]^{\mathrm{T}}$ *of a factorized exponential family, where all the* $\boldsymbol{T}_i(s_t^i)$ *have dimension larger or equal to 2, and b) the output* $\boldsymbol{T}_{NN}(\mathbf{s}_t)$ *of a neural network with ReLU activations. The resulting density function is thus given by*

$$p_{\boldsymbol{T}, \boldsymbol{\lambda}}(\mathbf{s}_t | \mathbf{a}_t, \mathbf{e}) = \mathcal{Q}(\mathbf{s}_t) / \mathcal{Z}(\mathbf{a}_t, \mathbf{e}) \exp\left[\boldsymbol{T}(\mathbf{s}_t)^{\mathrm{T}} \boldsymbol{\lambda}(\mathbf{a}_t, \mathbf{e})\right], \quad (2)$$

*where* $\mathcal{Q}$ *is the base measure and* $\mathcal{Z}$ *the normalizing constant.*

A neural network with ReLU activation has universal approximation power. Therefore, the term $\boldsymbol{T}_{NN}(\mathbf{s}_t)$ in Assumption 3 will allow the prior to capture arbitrary dependencies between the latent variables $\mathbf{s}_t$. This is important because for any two $s_t^i$ and $s_t^j$, we usually have that $s_t^i \not\perp\!\!\!\perp s_t^j | \mathbf{a}_t, \mathbf{e}$. The conditional dependencies between them are mainly due to the confounder $\mathbf{c}_{t-1}$ (cf. Fig. 1) (de Haan et al., 2019). Another possibility is that there might exist instantaneous effects between $s_t^i$ and $s_t^j$ (i.e., an arrow between $s_t^i$ and $s_t^j$) in some scenarios (Peters et al., 2017; Sutton & Barto, 2018), which is not considered in previous work (Bica et al., 2021).

Now we are ready to address the policy generalization problem in IL, by following the three phases in iCaRL described in Appendix C.1. For ease of reference, this approach is called *iCaRL-IL*, which is described in Appendix F.3 and whose corresponding theoretical results are given in Appendix H.

## 3 GENERALIZATION IN REINFORCEMENT LEARNING

### 3.1 PROBLEM FORMULATION

Unless stated otherwise, we follow the same notations used in the IL setting as described in Section 2.1. In the RL setting[2], we consider three generalization problems: policy generalization, representation

---

[2]We briefly review the basics of RL in Appendix A.

generalization, and dynamics generalization. As analysed in Section 1, they all can be resolved in the iCaRL framework. Specifically, we first identify the true states $\mathbf{s}_t$ from observations $\mathbf{x}_t$ (Phase 1), then discover the causal ancestors of rewards or the true causes of each state variable among the identified states $\mathbf{s}_t$ (Phase 2), and finally base on them to build invariant predictors in terms of policy, representation, and dynamics (Phase 3). Apparently, all the three generalization problems are same in Phase 1 but not in Phases 2&3. Hence, we first describe what they have in common in Phase 1 (Section 3.1) and then separately describe Phases 2&3 for each (Sections 3.2,3.3&3.4).

### 3.1.1 Assumptions on the Causal Diagram

Unlike in the IL setting where the reward is not given, we assume here without loss of generality that $\mathbf{s}_t^p$ and $\mathbf{s}_t^c$ are multiple scalar *causal factors* and *non-causal factors* of $r_{t+1}$. We also assume that all the environment $\mathcal{E}_{all}$ share some latent structure as encapsulated in Fig. 4 of the appendix and summarized below. Note that, since $\mathbf{a}_t$ is always a causal factor of $r_{t+1}$, it is not placed in the figure for simplicity.

**Assumption 4.** *(a) The causal graph containing $\mathbf{s}_t$ and $r_{t+1}$ is a DAG; (b) $\mathbf{x}_t \perp\!\!\!\perp r_{t+1}, \mathbf{e}|\mathbf{s}_t$, implying that $p(\mathbf{x}_t|\mathbf{s}_t)$ is invariant across $\mathcal{E}_{all}$.*

The practicality of Assumption 4 can be detailedly explained in a similar way to that of Assumption 2. A more in-depth explanation can be found in Appendix G.1.

### 3.1.2 Assumptions on the Prior

When the underlying causal diagram across $\mathcal{E}_{all}$ satisfies Assumption 4b, it is straightforward to obtain the primary assumption (Assumption 12 in Appendix E) over the prior $p(\mathbf{s}_t|r_{t+1}, \mathbf{e})$ leading to identifiability of the latent variables $\mathbf{s}_t$ by substituting $\mathbf{a}_t$ with $r_{t+1}$ in Assumption 3.

Under Assumptions 4b&12, we can follow the exact same steps as described in Appendix F.3.1 to identify the latent variables $\mathbf{s}_t$ with similar theoretical guarantees on identifiability, by directly substituting $\mathbf{a}_t$ with $r_{t+1}$ in all the corresponding equations and theorems. This resulting NF-iVAE model in the RL setting is called *NF-iVAE-RL*.

## 3.2 Policy Generalization

After estimating $\mathbf{s}_t$ for $\mathbf{x}_t$ in Phase 1, let us first consider the ideal generalization case in which an optimal policy learned from multiple training environments can zero-shot generalize to unseen testing environments. As discussed in Section 1, it has proved that a representation only consisting of all the causal ancestors of the reward is the minimal sufficient representation (MSR) for policy learning. Hence, in Phase 2 we need to discover the causal ancestors of the reward, denoted by $\mathbf{s}_t^A \equiv \mathbf{AN}(R)$, from the identified state $\mathbf{s}_t$. Due to Markovianity of the dynamics in RL, this can be implemented by recursively applying the two-step method described in Appendix F.3.2. Precisely, we first use the method to discover the direct causes $\mathbf{s}_t^p$ of the reward $r_{t+1}$ from $\mathbf{s}_t$ and then reuse it to find the direct causes of $\mathbf{s}_{t+1}^p$ from $\mathbf{s}_t$, whose union produces $\mathbf{s}_t^A$. After obtaining $\mathbf{s}_t^A$, to learn an invariant policy that generalizes well to unseen environments in Phase 3, we need to further assume that all the environments $\mathcal{E}_{all}$ share the following latent structure:

**Assumption 5.** *(a) $p(r_{t+1}|\mathbf{s}_t^p, \mathbf{a}_t)$ is invariant across $\mathcal{E}_{all}$; (b) $p(\mathbf{s}_{t+1}^A|\mathbf{s}_t^A, \mathbf{a}_t)$ is invariant across $\mathcal{E}_{all}$.*

Assumption 5a states that all the environments share the same reward function. Assumption 5b means that the latent structure induced by MSR $\mathbf{s}_t^A$ is shared across $\mathcal{E}_{all}$. Thus, it is straightforward that under Assumption 5, the learned policy across $\mathcal{E}_{tr}$ is guaranteed to generalize well to $\mathcal{E}_{all}$. We therefore have the following result with proof in Appendix H.

**Proposition 1.** *Under Assumptions 4,12&5 and the assumptions of Theorems 7&8, the policy learned across $\mathcal{E}_{tr}$ in the limit of infinite data has optimal OOD generalization across $\mathcal{E}_{all}$.*

Note that, unlike the IL setting where we learn the invariant policy in the supervised way, in the RL setting we follow Zhang et al. (2020b) and combine our MSR $\mathbf{s}_t^A$ with the soft actor-critic (SAC) algorithm (Haarnoja et al., 2018) to devise a practical RL method, termed *iCaRL-RL-P*, as shown in Algorithm 1 of the appendix (see more in Appendix G.3).

## 3.3 Representation Generalization

When Assumption 5 is violated, it is theoretically impossible that a policy learned from $\mathcal{E}_{tr}$ can generalize well to $\mathcal{E}_{all}$. As discussed in Section 1, however, it is still possible that the representation function $\Phi$ learned from $\mathcal{E}_{tr}$ can generalize to unseen testing environments $\mathcal{E}_{te}$, as long as the new reward functions across $\mathcal{E}_{te}$ are causally dependent on a subset of the same causal ancestors $\mathbf{s}^A$ that determine the original reward function across $\mathcal{E}_{tr}$. This condition can be formalized as follows:

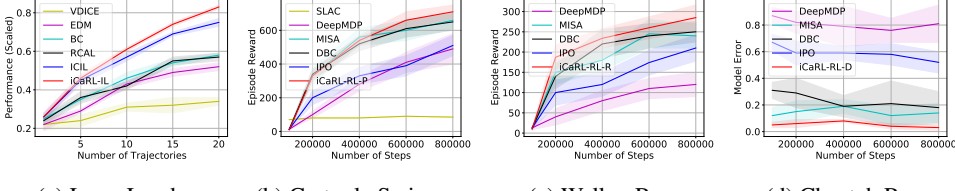

(a) LunarLander    (b) Cartpole Swingup    (c) Walker Run    (d) Cheetah Run

Figure 2: (a) Evaluation for policy generalization on OpenAI gym environments in the IL setting. $x$-axis indicates the number of trajectories with expert demonstrations from each training environment and $y$-axis is average return of the learned imitation policy on the test environment, scaled between 1 (expert performance) and 0 (random policy performance). (b-d) Evaluation for policy, representation, and dynamics generalization in the RL setting, where the policies, representations, and dynamics are trained on two environments with different simple distractors and evaluated on an unseen environment with natural video distractors. Note that, here the baselines are BC (Pomerleau, 1991), RCAL (Piot et al., 2014), VDICE (Kostrikov et al., 2019), EDM (Jarrett et al., 2020), IRM (Arjovsky et al., 2019), ICIL (Bica et al., 2021), SLAC (Lee et al., 2020), DeepMDP (Gelada et al., 2019), MISA (Zhang et al., 2020a), DBC (Zhang et al., 2020b), and IPO (Sonar et al., 2021).

**Assumption 6.** $\mathbf{AN}(R^{e^*}) = \mathbf{s}^A$ for any $e^* \in \mathcal{E}_{te}$.

Then, we have the following result whose proof is in Appendix H.

**Proposition 2.** *Under Assumptions 4,12&6 and the assumptions of Theorems 7&8, the representation function $\Phi$ learned across $\mathcal{E}_{tr}$ in the limit of infinite data is able to generalize to $\mathcal{E}_{te}$.*

### 3.4 DYNAMICS GENERALIZATION

When Assumption 6 is also not satisfied, then generalizing $\Phi$ is hopeless as well. However, it is still possible to learn generalizable (local) dynamics models by building invariant predictors on a per-state-variable level. As discussed in Section 1, the rationale behind is that in many world scenarios, each state variable only depends on a small subset of those state variables in the previous timestep, which is summarized as below.

**Assumption 7.** *Given states $\mathbf{s}_t, \mathbf{s}_{t+1}$ and action $\mathbf{a}_t$, we have $p(\mathbf{s}_{t+1}|\mathbf{s}_t, \mathbf{a}_t) = \prod_i p(s_{t+1}^i | \mathbf{s}_t^{P_i}, \mathbf{a}_t)$, where $s_{t+1}^i$ denotes the $i$-th dimension of state $\mathbf{s}_{t+1}$ and $\mathbf{s}_t^{P_i}$ is a set of the parents of $s_{t+1}^i$ in $\mathbf{s}_t$.*

To learn invariant (local) dynamics models that generalize well to unseen environments, we need to further assume that all the environments $\mathcal{E}_{all}$ share the following latent structure:

**Assumption 8.** *Given states $\mathbf{s}_t, \mathbf{s}_{t+1}$ and action $\mathbf{a}_t$, there exists some (local) dynamics model $p(s_{t+1}^i | \mathbf{s}_t^{P_i}, \mathbf{a}_t)$ that is invariant across $\mathcal{E}_{all}$.*

Graphically, Assumption 8 indicates that $s_{t+1}^i \perp\!\!\!\perp \mathbf{e}|\mathbf{s}_t^{P_i}$, as shown in Fig. 5 of the appendix. Without loss of generality, we similarly assume that $\mathbf{s}_t^{P_i} \doteq (s_t^{p_1^i}, \ldots, s_t^{p_r^i})$ and $\mathbf{s}_t^{C_i} \doteq (s_t^{c_1^i}, \ldots, s_t^{c_k^i})$ are multiple scalar *causal factors* and *non-causal factors* of $s_{t+1}^i$. After estimating $\mathbf{s}_t$ for $\mathbf{x}_t$ in Phase 1, it is straightforward to use the two-step approach described in Appendix F.3.2 to discover the direct causes $\mathbf{s}_t^{P_i}$ of $s_{t+1}^i$ from $\mathbf{s}_t$. Then, under Assumption 8, we can follow the same step depicted in Appendix F.3.3 to learn the invariant (local) dynamics model $p(s_{t+1}^i | \mathbf{s}_t^{P_i}, \mathbf{a}_t)$ in Phase 3 by directly substituting $\mathbf{s}_t^p$ with $(\mathbf{s}_t^{P_i}, \mathbf{a}_t)$ and $\mathbf{a}_t$ with $s_{t+1}^i$ in Eq. (20). This approach is called *iCaRL-RL-D*. We further have the following result (proof in Appendix H).

**Proposition 3.** *Under Assumptions 4,12,7&8 and the assumptions of Theorems 7&8, the (local) dynamics learned across $\mathcal{E}_{tr}$ in the limit of infinite data has optimal OOD generalization across $\mathcal{E}_{all}$.*

## 4 EXPERIMENTS

We compare our approach with a variety of baselines on both synthetic and real-world datasets. In all comparisons, unless stated otherwise, we average performance over ten runs and show the mean results with standard deviations. Due to space limit, we only highlight some key results as shown in Fig. 2 on the widely used control tasks and refer readers to Appendix I for more details. Our methods consistently attain improved generalization over a variety of baselines in both IL and RL settings.

## 5 CONCLUSION AND RELATED WORK

We investigated the different generalization problems in terms of policy, representation, and dynamics in IL and RL from a unified view, proposing a framework to tackle them with theoretical guarantees on both identifiability and generalizability under mild assumptions over environmental changes. Experimental results show that our methods attain improved generalization over a variety of baselines.

In Appendix B, we discuss the related work on different generalization problems in both IL and RL.

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

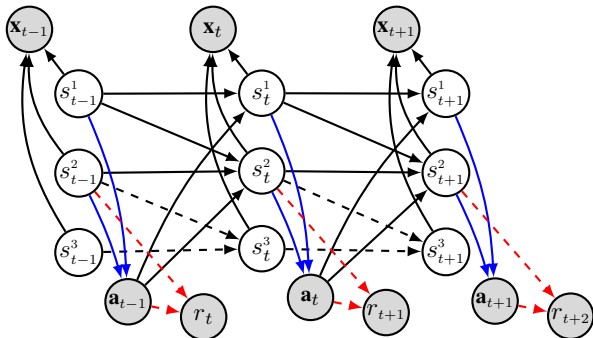

Figure 3: A graphical illustration of different generalization problems in IL and RL that we study in this work. Grey nodes denote observed variables and white nodes represent unobserved variables. Dashed lines denote the edges which might vary across environments and even be absent in some scenarios, whilst solid lines indicate that they are invariant across all the environments. In this toy example, we have state $\mathbf{s} = (s^1, s^2, s^3)$ and its corresponding observation $\mathbf{x}$, where $\mathbf{x}$ is assumed to contain all information about $\mathbf{s}$ (Assumption 1). In the IL settings where observation-action pairs $(\mathbf{x}, \mathbf{a})$ are collected from an expert policy (blue lines), an invariant policy should solely depend on $(s^1, s^2)$ extracted from $\mathbf{x}$. In the RL settings, it is worth noting that while $s^2$ is the only causal parent of $r$ in $\mathbf{s}$, a MSR must include both $s^1$ and $s^2$ because the next-timestep distribution of $s^2$ depends on $s^1$. We have three different RL generalization problems. (1) Representation generalization: a learned representation function that extracts $(s^1, s^2)$ from $\mathbf{x}$ can generalize to unseen environments. (2) Policy generalization: a learned policy mapping from $(s^1, s^2)$ to $\mathbf{a}_t$ can generalize to unseen environments. (3) Dynamics generalization: a learned local dynamics model $p(s_{t+1}^2|s_t^1, s_t^2, \mathbf{a}_t)$ can generalize to unseen environments.

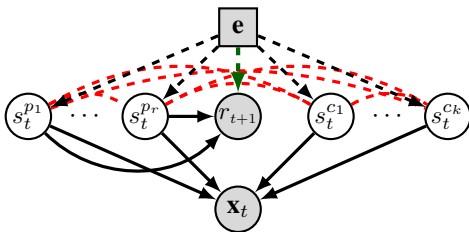

Figure 4: Causal structure of both policy and representation generalization in the RL settings, where we assume there exist multiple unobserved state variables. Each of them could be either a parent of $r_{t+1}$, or has no direct connection with $r_{t+1}$. We allow for arbitrary connections between the latent state variables (red dashed lines) as long as the resulting causal diagram including $r_{t+1}$ is a directed acyclic graph (DAG).

## A  BASICS OF IMITATION AND REINFORCEMENT LEARNING

We describe an environment as a standard Markov decision process (MDP) (Puterman, 1994; Sutton & Barto, 2018) given by a tuple $\mathcal{M} = (\mathcal{X}, \mathcal{A}, P, R, \gamma)$, with observations $\mathbf{x} \in \mathcal{X} \subseteq \mathbb{R}^d$, actions $\mathbf{a} \in \mathcal{A}$, a transition dynamics $P \equiv p(\mathbf{x}'|\mathbf{x}, \mathbf{a})$, a reward function $R(\mathbf{x}, \mathbf{a}) \in \mathbb{R}$, and a discount factor $\gamma \in [0, 1)$. A policy $\pi(\cdot|\mathbf{x})$ defines a distribution over actions conditioned on the observation $\mathbf{x}$. At time $t$, an agent is provided with an observation $\mathbf{x}_t \in \mathcal{X}$ and chooses an action $\mathbf{a}_t \in \mathcal{A}$ according to a policy $\mathbf{a}_t \sim \pi(\mathbf{x}_t)$. The agent receives a reward $r_{t+1} = R(\mathbf{x}_t, \mathbf{a}_t)$ and then the environment yields next observation $\mathbf{x}_{t+1} \sim p(\cdot|\mathbf{x}_t, \mathbf{a}_t)$. Under a policy $\pi$, the *value function*, denoted by $V_\pi(\mathbf{x})$, represents the expected cumulative discounted rewards (a.k.a., return) following $\pi$ from observation $\mathbf{x}$, defined by $V_\pi(\mathbf{x}) = \mathbb{E}_\pi[\sum_{t=0}^\infty \gamma^t r_{t+1}|\mathbf{x}_0 = \mathbf{x}]$. Similarly, we also can define the *action-value function* for policy $\pi$, denoted by $Q_\pi(\mathbf{x}, \mathbf{a})$, meaning that the value of taking action $\mathbf{a}$ in observation $\mathbf{x}$ under a policy $\pi$: $Q_\pi(\mathbf{x}, \mathbf{a}) = \mathbb{E}_\pi[\sum_{t=0}^\infty \gamma^t r_{t+1}|\mathbf{x}_0 = \mathbf{x}, \mathbf{a}_0 = \mathbf{a}]$. In the RL setting, the agent's goal is to find an optimal policy $\pi_*$ that maximizes the value function: $\pi_* = \arg\max_\pi V_\pi(\mathbf{x})$ for all $\mathbf{x} \in \mathcal{X}$.

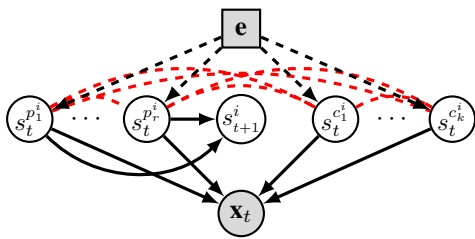

Figure 5: Causal structure of dynamics generalization in the RL settings, where we assume there exist multiple unobserved state variables. Each of them could be either a parent of $s_{t+1}^i$, or has no direct connection with $s_{t+1}^i$. We allow for arbitrary connections between the latent state variables (red dashed lines) as long as the resulting causal diagram including $s_{t+1}^i$ is a DAG.

In the IL setting, the primary difference from the previous RL setting is that the reward is neither known nor observed. Instead, we only have access to a dataset of expert demonstrations $\mathcal{D} = \{\tau_i\}_i^N$, where each demonstration $\tau$ consists of a sequence of observation-action pairs $\tau = (\mathbf{x}_t, \mathbf{a}_t)_{t=0,\dots}$ that are drawn from an expert policy $\pi_*$. We consider behavioural cloning (BC) approaches (Widrow, 1964; Pomerleau, 1989; 1991; Bain & Sammut, 1995; Schaal, 1999; Muller et al., 2006; Mülling et al., 2013; Bojarski et al., 2016; Mahler & Goldberg, 2017; Bansal et al., 2018) that reduce policy learning to supervised learning by training a discriminative model to predict expert actions given observations. That is, the goal of BC is to seek a policy $\pi$ that imitates the expert policy $\pi_*$ by solving the optimization problem:

$$\min_\pi \sum_{\tau \in \mathcal{D}} \sum_{(\mathbf{x}_t, \mathbf{a}_t) \in \tau} \ell(\pi(\mathbf{x}_t), \mathbf{a}_t), \tag{3}$$

where $\ell$ is a choice of loss function.

## B  RELATED WORK

**Generalization in IL.**  IL through behavioural cloning has been extensively studied (Widrow, 1964; Pomerleau, 1989; 1991; Bain & Sammut, 1995; Schaal, 1999; Muller et al., 2006; Kober et al., 2010; Bojarski et al., 2016; Mahler & Goldberg, 2017; Bansal et al., 2018). However, none of existing methods consider the problem of learning policies robust to spurious correlations so that they can generalize to unseen environments. There are also some work studying the problem of domain adaptation and transfer learning for IL. However, they assume access to either demonstrations from testing environments (Sermanet et al., 2017; Liu et al., 2018; Kim et al., 2020) or online interaction (de Haan et al., 2019; Lu & Tompson, 2020; Swamy et al., 2021), or they study the different problem of hidden confounding (de Haan et al., 2019; Etesami & Geiger, 2020; Zhang et al., 2020c; Kumor et al., 2021; Swamy et al., 2021). Another line of work is in the field of meta-learning whose goal is to generalize learned policies to new tasks, but they also require access to demonstrations from the new tasks (Finn et al., 2017a;b; Duan et al., 2017; Yu et al., 2018; James et al., 2018; Sharma et al., 2019). Perhaps the most related to ours is the one in Bica et al. (2021) that also tackles the generalization problem in IL by learning a representation shared across environments. However, they have no theoretical guarantees on both identifiability and generalizability. Also, their assumptions over the underlying causal graph are restricted, e.g., they assume that the causal factors $\mathbf{s}_t^p$ and non-causal factors $\mathbf{s}_t^c$ must be independent from each other, which is not necessary in ours; etc.

**Policy Generalization in RL.**  The goal of policy generalization in RL is to learn policies from multiple training environments so that they can zero-shot generalize to unseen testing environments. The most common approach to policy generalization is by adding different forms of regularization, such as dropout and batch normalization (Cobbe et al., 2019) and information-theoretic regularizer (Goyal et al., 2018; Pacelli & Majumdar, 2020). Although easy to implement, these methods do not explicitly exploit any (causal) structure of the RL problem. Data augmentation and domain randomization also show their potential in the sim-to-real generalization problem (Akkaya et al., 2019; Peng et al., 2018; Urakami et al., 2019), but they are complementary to our methods and could be potentially used to generate a diverse set of training environments for ours. While the approaches based on PAC-Bayes (Majumdar et al., 2021) and adversarial perturbations (Sinha et al., 2018) provide provable generalization guarantees, they require an *a priori* bound on how much the test environments differs from the training environments. Inspired by the work of Arjovsky et al.

(2019) and Ahuja et al. (2020), Krueger et al. (2021) propose a risk-extrapolation method and Sonar et al. (2021) propose an approach referred as to invariant policy optimization for OOD generalization in RL, but both have no theoretical guarantees on identifiability and generalizability. Saengkyongam et al. (2021) tackle the problem of environmental shifts in offline contextual bandits from a causal perspective. There are also much work on quantifying generalization in RL (Mnih et al., 2013; Nichol et al., 2018; Song et al., 2019; Cobbe et al., 2019; 2020; Ahmed et al., 2020; Kirk et al., 2021).

**Representation Generalization in RL.** This line of work focuses on representation generalization in RL that aims to learn representations from multiple training environments so that they can zero-shot generalize to unseen testing environments. Much work has been done on reconstruction-based representations (Lange & Riedmiller, 2010; Lange et al., 2012; Watter et al., 2015; Hafner et al., 2019; Gelada et al., 2019; Huang et al., 2021), contrastive-based representations (Oord et al., 2018; Chen et al., 2020; Laskin et al., 2020), and bisimulation-based representations (Larsen & Skou, 1989; Taylor et al., 2008; Ferns et al., 2011; Ferns & Precup, 2014; Castro, 2020). However, none of these methods explicitly consider the OOD generalization problem in terms of representation across environments. Zhang et al. (2020a) propose a method of invariant prediction to learn model-invariance state abstractions that generalize to novel observations in the multi-environment setting, but their method is limited to a family of environments represented by block MDPs (Du et al., 2019) and also has no theoretical guarantees on identifiability and generalizability. Lee et al. (2021) propose an approach for structure and transfer learning of robot manipulation policies, but it requires performing/simulating the effect of interventions in the environment. Agarwal et al. (2021) incorporate the inherent sequential structure in RL into the representation learning process to improve generalization. Zhang et al. (2020b) leverage bisimulation metrics to learn generalizable representations which encodes only the task-relevant information from observations. Similarly, both of the methods above provide no theoretical guarantees on identifiability and generalizability.

**Dynamics Generalization in RL.** For the tasks of dynamics generalization in RL, the goal is to learn (local) dynamics models from multiple training environments so that they can zero-shot generalize to unseen testing environments. Boutilier et al. (1999) detailedly explore structural assumptions and computational leverage on the underlying MDP for decision-theoretic planning. The most related setting is of factored MDPs, but most existing work in this field is either to assume a known causal structure for the transition dynamics (Kearns & Koller, 1999; Jonsson & Barto, 2006; Strehl et al., 2007; Osband & Van Roy, 2014) or to not learn states abstractions (Kearns & Koller, 1999; Strehl et al., 2007; Misra et al., 2020). Hallak et al. (2015) discuss learning the factored structure in the dynamics of the environment under the factored MDP assumption. Volodin et al. (2020) consider the problem of inferring a causal model of the environment, but intervention is required. Note that, none of these methods consider the generalization problem in terms of dynamics. Also note that, most factored MDP works include the factored reward condition, which is not necessary in ours. Another thing in which we differ from these methods is that we only focus on generalizing the local dynamics model on a per-state-variable level. The most related to our work is the one in Tomar et al. (2021). While attempting to learn generalizable local dynamics model, they have no theoretical guarantees on identifiability and generalizability.

## C  VARIATIONAL AUTOENCODERS

We briefly describe the framework of variational autoencoders (VAEs), which allows us to efficiently learn deep latent-variable models and their corresponding inference models (Kingma & Welling, 2013; Rezende et al., 2014). Consider a simple latent variable model where $\mathbf{x} \in \mathbb{R}^d$ stands for an observed variable and $\mathbf{z} \in \mathbb{R}^n$ for a latent variable. A VAE method learns a full generative model $p_{\boldsymbol{\theta}}(\mathbf{x}, \mathbf{z}) = p_{\boldsymbol{\theta}}(\mathbf{x}|\mathbf{z})p_{\boldsymbol{\theta}}(\mathbf{z})$ and an inference model $q_{\boldsymbol{\phi}}(\mathbf{z}|\mathbf{x})$, typically a factorized Gaussian distribution whose mean and variance parameters are given by the output of a neural network with input $\mathbf{x}$. This inference model approximates the posterior $p_{\boldsymbol{\theta}}(\mathbf{z}|\mathbf{x})$, where $\boldsymbol{\theta}$ is a vector of parameters of the generative model, $\boldsymbol{\phi}$ a vector of parameters of the inference model, and $p_{\boldsymbol{\theta}}(\mathbf{z})$ is a prior distribution over the latent variables. Instead of maximizing the data log-likelihood, we maximize its lower bound $\mathcal{L}_{\text{VAE}}(\boldsymbol{\theta}, \boldsymbol{\phi})$:

$$\log p_{\boldsymbol{\theta}}(\mathbf{x}) \geq \mathcal{L}_{\text{VAE}}(\boldsymbol{\theta}, \boldsymbol{\phi}) := \mathbb{E}_{q_{\boldsymbol{\phi}}(\mathbf{z}|\mathbf{x})}\left[\log p_{\boldsymbol{\theta}}(\mathbf{x}|\mathbf{z})\right] - \text{KL}\left(q_{\boldsymbol{\phi}}(\mathbf{z}|\mathbf{x})||p_{\boldsymbol{\theta}}(\mathbf{z})\right),$$

where we have used Jensen's inequality, and $\text{KL}(\cdot||\cdot)$ denotes the Kullback-Leibler divergence between two distributions.

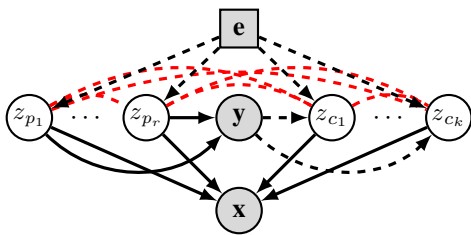

Figure 6: Causal structure of iCaRL, where we assume there exist multiple latent variables. Each of them could be a parent, a child of **y**, or have no direct connection with **y**. Note that iCaRL allows for arbitrary connections between the latent variables (red dashed lines) as long as the resulting causal diagram including **y** is a directed acyclic graph (DAG).

## C.1 INVARIANT CAUSAL REPRESENTATION LEARNING (ICARL)

Invariant causal representation learning (iCaRL) (Lu et al., 2022) is a framework for out-of-distribution (OOD) generalization in supervised learning (SL), with theoretical guarantees on both identifiability and generalizability. It aims to construct an invariant predictor that performs well across all environments $\mathcal{E}_{all}$ by exploiting data collected from multiple training environments $\mathcal{E}_{tr}$, where $\mathcal{E}_{tr} \subset \mathcal{E}_{all}$. The scenarios for OOD generalization in SL that iCaRL mainly considers are the ones whose underlying data generating process can be represented by a general causal diagram as shown in Fig. 6. In particular, the input data is $\mathbf{x} \in \mathbb{R}^d$ and its corresponding target is $\mathbf{y} \in \mathbb{R}^s$.[3] The data representation $\mathbf{z} = (z_{p_1}, \ldots, z_{p_r}, z_{c_1}, \ldots, z_{c_k}) \in \mathbb{R}^n$ are a set of latent variables encoding the input data, where $n = r + k$, and $\{z_i\}_{i \in I_p \doteq \{p_1, \ldots, p_r\}}$ and $\{z_j\}_{j \in I_c \doteq \{c_1, \ldots, c_k\}}$ are multiple scalar *causal factors* and *non-causal factors* of $\mathbf{y}$, respectively. We denote $\mathbf{z}_p \doteq (z_{p_1}, \ldots, z_{p_r})$ and $\mathbf{z}_c \doteq (z_{c_1}, \ldots, z_{c_k})$ for the ease of clarification. It is assumed that $\mathbf{z}$ is of lower dimension than $\mathbf{x}$, that is, $n \leq d$. The environment is treated as a random variable $\mathbf{e}$,[4] where $\mathbf{e}$ could be any information specific to the environment (Storkey, 2009; Peters et al., 2016; Zhang et al., 2017; Huang et al., 2020). For simplicity, $\mathbf{e}$ is set to be the environment index, i.e., $\mathbf{e} \in \{1, \ldots, N\}$, where $N$ is the number of training environments.

It is worth noting that iCaRL allows for arbitrary connections between the latent variables $\mathbf{z}$ as long as the resulting causal diagram including $\mathbf{y}$ is a directed acyclic graph (DAG). Dashed lines indicate the *causal mechanisms* which might vary across environments and even be absent in some scenarios, whilst solid lines indicate that they are invariant across all the environments. To sum up, we posit that the underlying causal graph (Fig. 6) satisfies the following assumption (Lu et al., 2022):

**Assumption 9.** *(a) The causal graph containing* $\mathbf{z}$ *and* $\mathbf{y}$ *is a DAG; (b)* $\mathbf{x} \perp\!\!\!\perp \mathbf{y}, \mathbf{e} | \mathbf{z}$, *implying that* $p(\mathbf{x}|\mathbf{z})$ *is invariant across* $\mathcal{E}_{all}$; *(c)* $\mathbf{y} \perp\!\!\!\perp \mathbf{e} | \mathbf{z}_p$, *implying that* $p(\mathbf{y}|\mathbf{z}_p)$ *is invariant across* $\mathcal{E}_{all}$.

To learn the invariant predictor $w$ that is a function mapping from $\mathbf{z}_p$ to $\mathbf{y}$, the iCaRL framework consists of three phases. The idea is to first identify the latent variables ($\mathbf{z}$) from the input data ($\mathbf{x}$) by using an extended identifiable variational autoencoder (NF-iVAE) model under Assumptions 9b&10 (Phase 1), then discover direct causes ($\mathbf{z}_p$) of the target ($\mathbf{y}$) among the identified latent variables under Assumption 9a (Phase 2), and finally learn the invariant predictor ($w$) for the target from the discovered causes under Assumption 9c (Phase 3). We briefly review them below, cf. also Lu et al. (2022).

## C.2 PHASE 1: IDENTIFYING LATENT VARIABLES

Technically, under Assumption 9b, the proposed NF-iVAE in Phase 1 is a conditional generative model with the parameters $\boldsymbol{\theta} = (\boldsymbol{f}, \boldsymbol{T}, \boldsymbol{\lambda}) \in \Theta$, defined as

$$p_{\boldsymbol{\theta}}(\mathbf{x}, \mathbf{z}|\mathbf{y}, \mathbf{e}) = p_{\boldsymbol{f}}(\mathbf{x}|\mathbf{z})p_{\boldsymbol{T}, \boldsymbol{\lambda}}(\mathbf{z}|\mathbf{y}, \mathbf{e}), \quad (4)$$

$$p_{\boldsymbol{f}}(\mathbf{x}|\mathbf{z}) = p_{\boldsymbol{\epsilon}}(\mathbf{x} - \boldsymbol{f}(\mathbf{z})), \quad (5)$$

where $\boldsymbol{\epsilon}$ is an independent noise variable with probability density function $p_{\boldsymbol{\epsilon}}(\boldsymbol{\epsilon})$. Importantly, the primary assumption leading to identifiability of the latent variables in NF-iVAE is that the conditional prior $p_{\boldsymbol{T}, \boldsymbol{\lambda}}(\mathbf{z}|\mathbf{y}, \mathbf{e})$ belongs to a general exponential family with sufficient statistics $\boldsymbol{T}$ and natural parameters $\boldsymbol{\lambda}$. This is formalized as follows:

---

[3] The setup applies to both continuous and categorical data. If any input data or target is categorical, we one-hot encode it.

[4] For generality, we use the bold font (**y** and **e**) to allow for multi-dimensional variables.

**Assumption 10.** $p_{T,\lambda}(\mathbf{z}|\mathbf{y}, \mathbf{e})$ *belongs to a general exponential family with parameter vector given by an arbitrary function $\boldsymbol{\lambda}(\mathbf{y}, \mathbf{e})$ and sufficient statistics $\boldsymbol{T}(\mathbf{z}) = [\boldsymbol{T}_f(\mathbf{z})^{\mathrm{T}}, \boldsymbol{T}_{NN}(\mathbf{z})^{\mathrm{T}}]^{\mathrm{T}}$ given by the concatenation of a) the sufficient statistics $\boldsymbol{T}_f(\mathbf{z}) = [\boldsymbol{T}_1(z_1)^{\mathrm{T}}, \cdots, \boldsymbol{T}_n(z_n)^{\mathrm{T}}]^{\mathrm{T}}$ of a factorized exponential family, where all the $\boldsymbol{T}_i(z_i)$ have dimension larger or equal to 2, and b) the output $\boldsymbol{T}_{NN}(\mathbf{z})$ of a neural network with ReLU activations. The resulting density function is thus given by*

$$p_{\boldsymbol{T},\boldsymbol{\lambda}}(\mathbf{z}|\mathbf{y}, \mathbf{e}) = \frac{\mathcal{Q}(\mathbf{z})}{\mathcal{Z}(\mathbf{y}, \mathbf{e})} \exp\left[\boldsymbol{T}(\mathbf{z})^{\mathrm{T}}\boldsymbol{\lambda}(\mathbf{y}, \mathbf{e})\right], \quad (6)$$

*where $\mathcal{Q}$ is the base measure and $\mathcal{Z}$ the normalizing constant.*

A neural network with ReLU activation has universal approximation power. Therefore, the term $\boldsymbol{T}_{NN}(\mathbf{z})$ in the above prior distribution will allow us to capture arbitrary dependencies between the latent variables.

Most importantly, Lu et al. (2022) prove that under Assumptions 9b&10 and the conditions stated in Theorems 1-3 of Lu et al. (2022), NF-iVAE can identify the latent variable $\mathbf{z}$.

### C.3 PHASE 2: DISCOVERING DIRECT CAUSES

After estimating $\mathbf{z}$ for each data point, under Assumption 9a, Lu et al. (2022) propose a heuristic two-step approach to discovering the direct causes $\mathbf{z}_p$ in Phase 2, which works well in practice. First, the PC algorithm (Spirtes et al., 2000) is conducted to learn a Markov equivalence class of DAGs, which outputs the direct neighbours of $\mathbf{y}$, denoted by $\mathbf{NE}(\mathbf{y})$. Second, all pairs[5] in $\mathbf{NE}(\mathbf{y})$ are tested with conditional independence testing (Zhang et al., 2012) to discover $\mathbf{z}_p$ by comparing $p$-values from the two tests: $\texttt{IndTest}(z_i, z_j|\mathbf{e})$ and $\texttt{IndTest}(z_i, z_j|\mathbf{y}, \mathbf{e})$, where $\texttt{IndTest}$ denotes (conditional) independence test. This is based on the observation that under Assumption 9, for any two $z_i$ and $z_j$ from $\mathbf{NE}(\mathbf{y})$, only when both are causes of $\mathbf{y}$ does the dependency between them generally increase after additionally conditioning on $\mathbf{y}$.

### C.4 PHASE 3: LEARNING AN INVARIANT PREDICTOR

After having obtained $\mathbf{z}_p$ for $\mathbf{y}$ across $\mathcal{E}_{tr}$, under Assumption 9c, the invariant predictor $w$ can be learned in Phase 3 by solving the following optimization problem:

$$\min_w \sum_{e \in \mathcal{E}_{tr}} \mathbb{E}_{\mathbf{z}_p^e, \mathbf{y}^e} \left[\ell(w(\mathbf{z}_p^e), \mathbf{y}^e)\right], \quad (7)$$

where $\ell(\cdot)$ could be any loss. Since $\mathbb{E}[\mathbf{y}|\mathbf{z}_p]$ is assumed to be invariant across $\mathcal{E}_{all}$, the learned $w$ is guaranteed to perform well across $\mathcal{E}_{all}$.

When in an unseen testing environment, $\mathbf{z}_p$ is required to be inferred from $\mathbf{x}$ before applying $w$ for prediction. This is implemented by leveraging the assumed invariant $p_{\boldsymbol{f}}(\mathbf{x}|\mathbf{z})$ through solving the following optimization problem:

$$\max_{\mathbf{z}_p, \mathbf{z}_c} \log p_{\boldsymbol{f}}(\mathbf{x}|\mathbf{z}_p, \mathbf{z}_c) + \lambda_1 ||\mathbf{z}_p||_2^2 + \lambda_2 ||\mathbf{z}_c||_2^2, \quad (8)$$

where the hyperparameters $\lambda_1 > 0$ and $\lambda_2 > 0$ control that the learned $\mathbf{z}_p$ and $\mathbf{z}_c$ have a reasonable scale, both of which are selected on training/validation data.

## D DEFINITIONS

For convenience, we restates some definitions given in Lu et al. (2022).

**Definition 1.** *A structural equation model (SEM) $\mathcal{C} := (\mathcal{S}, N)$ governing the random vector $\boldsymbol{X} = (X_1, \ldots, X_d)$ is a set of structural equations:*

$$\mathcal{S}_i : X_i \leftarrow f_i(\mathrm{Pa}(X_i), N_i),$$

*where $\mathrm{Pa}(X_i) \subseteq \{X_1, \ldots, X_d\} \setminus \{X_i\}$ are called the parents of $X_i$, and the $N_i$ are independent noise random variables. We say that "$X_i$ causes $X_j$" if $X_i \in \mathrm{Pa}(X_j)$. We call causal graph of $\boldsymbol{X}$ to the graph obtained by drawing i) one node for each $X_i$, and ii) one edge from $X_i$ to $X_j$ if $X_i \in \mathrm{Pa}(X_j)$. We assume acyclic causal graphs.*

---

[5]We only need to consider those variables in $\mathbf{NE}(\mathbf{y})$ whose edges connecting to $\mathbf{y}$ are not oriented by PC.

**Definition 2.** *Consider a SEM $\mathcal{C} := (\mathcal{S}, N)$. An intervention $e$ on $\mathcal{C}$ consists of replacing one or several of its structural equations to obtain an intervened SEM $\mathcal{C}^e := (\mathcal{S}^e, N^e)$, with structural equations:*

$$\mathcal{S}_i^e : X_i^e \leftarrow f_i^e(\mathrm{Pa}^e(X_i^e), N_i^e),$$

*The variable $\boldsymbol{X}^e$ is intervened if $\mathcal{S}_i \neq \mathcal{S}_i^e$ or $N_i \neq N_i^e$.*

**Definition 3.** *Consider a structural equation model (SEM) $\mathcal{S}$ governing the random vector $(X_1, \ldots, X_n, \boldsymbol{Y})$, and the learning goal of predicting $\boldsymbol{Y}$ from $\boldsymbol{X}$. Then, the set of all environments $\mathcal{E}_{all}(\mathcal{S})$ indexes all the interventional distributions $P(\boldsymbol{X}^e, \boldsymbol{Y}^e)$ obtainable by valid interventions $e$. An intervention $e \in \mathcal{E}_{all}(\mathcal{S})$ is valid as long as (i) the causal graph remains acyclic, (ii) $\mathbb{E}\left[\boldsymbol{Y}^e|\mathrm{Pa}(\boldsymbol{Y})\right] = \mathbb{E}\left[\boldsymbol{Y}|\mathrm{Pa}(\boldsymbol{Y})\right]$, and (iii) $\mathbb{V}\left[\boldsymbol{Y}^e|\mathrm{Pa}(\boldsymbol{Y})\right]$ remains within a finite range.*

**Definition 4.** *(**Exponential family**) A multivariate exponential family is a set of distributions whose probability density function can be written as*

$$p(\boldsymbol{X}) = \frac{\mathcal{Q}(\boldsymbol{X})}{\mathcal{Z}(\boldsymbol{\theta})} \exp(\langle \boldsymbol{T}(\boldsymbol{X}), \boldsymbol{\theta} \rangle), \tag{9}$$

*where $\mathcal{Q} : \mathcal{X} \to \mathbb{R}$ is the base measure, $\mathcal{Z}(\boldsymbol{\theta})$ is the normalizing constant, $\boldsymbol{T} : \mathcal{X} \to \mathbb{R}^k$ is the sufficient statistics, and $\boldsymbol{\theta} \in \mathbb{R}^k$ is the natural parameter. The size $k \geq n$ is the dimension of the sufficient statistics $\boldsymbol{T}$ and depends on the latent space dimension $n$. Note that $k$ is fixed given $n$.*

**Definition 5.** *(**Strongly exponential distributions**) A multivariate exponential family distribution*

$$p(\boldsymbol{X}) = \frac{\mathcal{Q}(\boldsymbol{X})}{\mathcal{Z}(\boldsymbol{\theta})} \exp(\langle \boldsymbol{T}(\boldsymbol{X}), \boldsymbol{\theta} \rangle) \tag{10}$$

*is strongly exponential, if*

$$(\exists \boldsymbol{\theta} \in \mathbb{R}^k \ s.t. \ \langle \boldsymbol{T}(\boldsymbol{X}), \boldsymbol{\theta} \rangle = const, \ \forall \boldsymbol{X} \in \mathcal{X}) \implies (l(\mathcal{X}) = 0 \ or \ \boldsymbol{\theta} = \boldsymbol{0}), \quad \forall \mathcal{X} \subset \mathbb{R}^n, \tag{11}$$

*where $l$ is the Lebesgue measure.*

The density of a strongly exponential distribution has almost surely the exponential component and can only be reduced to the base measure on a set of measure zero. Note that all common multivariate exponential family distributions (e.g. multivariate Gaussian) are strongly exponential.

**Definition 6.** *Let $\Theta$ be the domain of the parameters $\boldsymbol{\theta} = \{\boldsymbol{f}, \boldsymbol{T}, \boldsymbol{\lambda}\}$. Let $\sim$ be an equivalence relation on $\Theta$. A deep generative model is said to be $\sim$–identifiable if*

$$p_{\boldsymbol{\theta}}(\boldsymbol{O}) = p_{\tilde{\boldsymbol{\theta}}}(\boldsymbol{O}) \implies \boldsymbol{\theta} \sim \tilde{\boldsymbol{\theta}}. \tag{12}$$

*The elements in the quotient space $\Theta \backslash \sim$ are called the identifiability classes.*

**Definition 7.** *Let $\sim_A$ be an equivalence relation on $\Theta$ defined by:*

$$(\boldsymbol{f}, \boldsymbol{T}, \boldsymbol{\lambda}) \sim_A (\tilde{\boldsymbol{f}}, \tilde{\boldsymbol{T}}, \tilde{\boldsymbol{\lambda}}) \iff \exists A, \boldsymbol{c} \ s.t. \ \boldsymbol{T}(\boldsymbol{f}^{-1}(\boldsymbol{O})) = A\tilde{\boldsymbol{T}}(\tilde{\boldsymbol{f}}^{-1}(\boldsymbol{O})) + \boldsymbol{c}, \ \forall \boldsymbol{O} \in \mathcal{O}, \tag{13}$$

*where $A \in \mathbb{R}^{k \times k}$ is an invertible matrix and $\boldsymbol{c} \in \mathbb{R}^k$ is a vector.*

**Definition 8.** *Let $\sim_P$ be an equivalence relation on $\Theta$ defined by:*

$$(\boldsymbol{f}, \boldsymbol{T}, \boldsymbol{\lambda}) \sim_P (\tilde{\boldsymbol{f}}, \tilde{\boldsymbol{T}}, \tilde{\boldsymbol{\lambda}}) \iff \exists P, \boldsymbol{c} \ s.t. \ \boldsymbol{T}(\boldsymbol{f}^{-1}(\boldsymbol{O})) = P\tilde{\boldsymbol{T}}(\tilde{\boldsymbol{f}}^{-1}(\boldsymbol{O})) + \boldsymbol{c}, \ \forall \boldsymbol{O} \in \mathcal{O}, \tag{14}$$

*where $P \in \mathbb{R}^{k \times k}$ is a block permutation matrix and $\boldsymbol{c} \in \mathbb{R}^k$ is a vector.*

# E  ASSUMPTIONS

**Assumption 11.** *$p_{\boldsymbol{T}, \boldsymbol{\lambda}}(\mathbf{s}_t|\mathbf{a}_t, \mathbf{c}_{c-1}, \mathbf{e})$ belongs to a general exponential family with parameter vector given by an arbitrary function $\boldsymbol{\lambda}(\mathbf{a}_t, \mathbf{c}_{c-1}, \mathbf{e})$ and sufficient statistics $\boldsymbol{T}(\mathbf{s}_t) = [\boldsymbol{T}_f(\mathbf{s}_t)^{\mathrm{T}}, \boldsymbol{T}_{NN}(\mathbf{s}_t)^{\mathrm{T}}]^{\mathrm{T}}$ given by the concatenation of a) the sufficient statistics $\boldsymbol{T}_f(\mathbf{s}_t) = [\boldsymbol{T}_1(s_t^1)^{\mathrm{T}}, \cdots, \boldsymbol{T}_n(s_t^n)^{\mathrm{T}}]^{\mathrm{T}}$ of a factorized exponential family, where all the $\boldsymbol{T}_i(s_t^i)$ have dimension larger or equal to 2, and b) the output $\boldsymbol{T}_{NN}(\mathbf{s}_t)$ of a neural network with ReLU activations. The resulting density function is thus given by*

$$p_{\boldsymbol{T}, \boldsymbol{\lambda}}(\mathbf{s}_t|\mathbf{a}_t, \mathbf{c}_{c-1}, \mathbf{e}) = \frac{\mathcal{Q}(\mathbf{s}_t)}{\mathcal{Z}(\mathbf{a}_t, \mathbf{c}_{c-1}, \mathbf{e})} \exp\left[\boldsymbol{T}(\mathbf{s}_t)^{\mathrm{T}} \boldsymbol{\lambda}(\mathbf{a}_t, \mathbf{c}_{c-1}, \mathbf{e})\right], \tag{15}$$

*where $\mathcal{Q}$ is the base measure and $\mathcal{Z}$ the normalizing constant.*

**Assumption 12.** *$p_{\boldsymbol{T}, \boldsymbol{\lambda}}(\mathbf{s}_t|r_{t+1}, \mathbf{e})$ belongs to a general exponential family with parameter vector given by an arbitrary function $\boldsymbol{\lambda}(r_{t+1}, \mathbf{e})$ and sufficient statistics $\boldsymbol{T}(\mathbf{s}_t) = [\boldsymbol{T}_f(\mathbf{s}_t)^{\mathrm{T}}, \boldsymbol{T}_{NN}(\mathbf{s}_t)^{\mathrm{T}}]^{\mathrm{T}}$ given by the concatenation of a) the sufficient statistics $\boldsymbol{T}_f(\mathbf{s}_t) = [\boldsymbol{T}_1(s_t^1)^{\mathrm{T}}, \cdots, \boldsymbol{T}_n(s_t^n)^{\mathrm{T}}]^{\mathrm{T}}$ of a factorized exponential family, where all the $\boldsymbol{T}_i(s_t^i)$ have dimension larger or equal to 2, and b) the*

*output $\boldsymbol{T}_{NN}(\mathbf{s}_t)$ of a neural network with ReLU activations. The resulting density function is thus given by*

$$p_{\boldsymbol{T},\boldsymbol{\lambda}}(\mathbf{s}_t|r_{t+1},\mathbf{e}) = \frac{\mathcal{Q}(\mathbf{s}_t)}{\mathcal{Z}(r_{t+1},\mathbf{e})} \exp\left[\boldsymbol{T}(\mathbf{s}_t)^{\mathrm{T}}\boldsymbol{\lambda}(r_{t+1},\mathbf{e})\right],$$

*where $\mathcal{Q}$ is the base measure and $\mathcal{Z}$ the normalizing constant.*

# F    GENERALIZATION IN IMITATION LEARNING

## F.1    THE CONFOUNDER $\mathbf{c}_{t-1}$

Note that, as pointed out by de Haan et al. (2019), the state-action pair $\mathbf{c}_{t-1} = (\mathbf{s}_{t-1}, \mathbf{a}_{t-1})$ in the previous timestep serves as a confounder influencing each state variable in $\mathbf{s}_t$. Hence, we can explicitly consider it in the causal diagram, as shown in Fig. 1. In practice, we use the observation-action pair $(\mathbf{x}_{t-1}, \mathbf{a}_{t-1})$ to approximate $\mathbf{c}_{t-1}$ (Assumption 1) due to the unobservability of $\mathbf{s}_{t-1}$. Thus, $\mathbf{c}_{t-1}$ can be thought of as an observed confounder. In NF-iVAE, such an observed $\mathbf{c}_{t-1}$ can be treated as an additional auxiliary variable other than $\mathbf{a}_t$ and $\mathbf{e}$, which provides more information leading to identifiability. That is, in some scenarios where the additional information of $(\mathbf{a}_t, \mathbf{e})$ is not enough to identify $\mathbf{s}_t$, we can include $\mathbf{c}_{t-1}$ to help further identify $\mathbf{s}_t$ with theoretical guarantees. This can be done by directly substituting $(\mathbf{a}_t, \mathbf{e})$ with $(\mathbf{c}_{t-1}, \mathbf{a}_t, \mathbf{e})$ in the previous assumptions, equations, and theorems (Appendix F.4 & H). This technique also applies to the different generalization problems in RL that we will describe below. For simplicity, we will no longer include the node $\mathbf{c}_{t-1}$ in the subsequent causal diagrams.

## F.2    EXPLAINING THE PRACTICALITY OF ASSUMPTION 2

Let us explain in more detail how practical Assumption 2 is in the IL setting. Assumption 2a is a common assumption in causal discovery (Spirtes et al., 2000; Pearl, 2009; Peters et al., 2017). The reason that we require it is that we need to leverage causal discovery algorithms to discover the causes of expert action $\mathbf{a}_t$ among $\mathbf{s}_t$ in Phase 2. It also makes sense in Assumption 2b that the generative/causal mechanism $p(\mathbf{x}_t|\mathbf{s}_t)$ is assumed to be invariant across $\mathcal{E}_{all}$. First, such a causal mechanism mapping latent variables to observations is widely adopted in the machine learning literature (Thrun et al., 2005; Hyvarinen & Morioka, 2016; Suter et al., 2019; Hyvärinen et al., 2019; Teshima et al., 2020; Khemakhem et al., 2020; Schölkopf et al., 2021; Lu et al., 2022; Sun et al., 2021). Second, we can view the generative process $p(\mathbf{x}_t|\mathbf{s}_t)$ as a physical mechanism generating the observed variables, which is reasonably stable across $\mathcal{E}_{all}$ (Peters et al., 2017; Schölkopf, 2019; Ahmed et al., 2020; Schölkopf et al., 2021). Assumption 2c is a widely-used default assumption in OOD generalization (Peters et al., 2016; Arjovsky et al., 2019) and is also what we aim for in the IL setting. Apparently, Assumption 2, together with its corresponding causal diagram in Fig. 1, is flexible enough to cover most scenarios in the IL setting.

## F.3    POLICY GENERALIZATION

Now we are ready to address the policy generalization problem in IL, by following the three phases in iCaRL described in Appendix C.1. For ease of reference, this approach is called *iCaRL-IL*.

### F.3.1    PHASE 1: IDENTIFYING STATES

Under Assumption 2b, it is straightforward to obtain a corresponding generative model by directly substituting $(\mathbf{x}, \mathbf{z}, \mathbf{y})$ with $(\mathbf{x}_t, \mathbf{s}_t, \mathbf{a}_t)$ in Eqs. (4-5):

$$p_{\boldsymbol{\theta}}(\mathbf{x}_t, \mathbf{s}_t|\mathbf{a}_t, \mathbf{e}) = p_{\boldsymbol{f}}(\mathbf{x}_t|\mathbf{s}_t)p_{\boldsymbol{T},\boldsymbol{\lambda}}(\mathbf{s}_t|\mathbf{a}_t, \mathbf{e}), \tag{16}$$

$$p_{\boldsymbol{f}}(\mathbf{x}_t|\mathbf{s}_t) = p_{\boldsymbol{\epsilon}}(\mathbf{x}_t - \boldsymbol{f}(\mathbf{s}_t)). \tag{17}$$

The corresponding evidence lower bound (ELBO) is

$$\mathcal{L}_{\text{phase1}}^{\text{ELBO}}(\boldsymbol{\theta},\boldsymbol{\phi}) := \mathbb{E}_{\mathcal{D}_{\mathcal{E}_{tr}}}\left[\mathbb{E}_{q_{\boldsymbol{\phi}}(\mathbf{s}_t|\mathbf{x}_t,\mathbf{a}_t,\mathbf{e})}\left[\log p_{\boldsymbol{f}}(\mathbf{x}_t|\mathbf{s}_t) + \log p_{\boldsymbol{T},\boldsymbol{\lambda}}(\mathbf{s}_t|\mathbf{a}_t,\mathbf{e}) - \log q_{\boldsymbol{\phi}}(\mathbf{s}_t|\mathbf{x}_t,\mathbf{a}_t,\mathbf{e})\right]\right], \tag{18}$$

where $q_{\boldsymbol{\phi}}(\mathbf{s}_t|\mathbf{x}_t,\mathbf{a}_t,\mathbf{e})$ is an approximate conditional distribution for $\mathbf{s}_t$ given by a recognition network with parameters $\boldsymbol{\phi}$. To guarantee the identifiability result, the prior $p_{\boldsymbol{T},\boldsymbol{\lambda}}(\mathbf{s}_t|\mathbf{a}_t,\mathbf{e})$ is assumed to satisfy Assumption 3 (i.e. Eq. (15)). Considering the unknown normalization constant $\mathcal{Z}$ in this prior, we

use score matching (Hyvärinen, 2005; Vincent, 2011) to learn $(\boldsymbol{T}, \boldsymbol{\lambda})$ by minimizing

$$\mathcal{L}_{\text{phase1}}^{\text{SM}}(\boldsymbol{T}, \boldsymbol{\lambda}) := \mathbb{E}_{\mathcal{D}_{\mathcal{E}_{tr}}} \left[ \mathbb{E}_{q_{\boldsymbol{\phi}}(\mathbf{s}_t | \mathbf{x}_t, \mathbf{a}_t, \mathbf{e})} \left[ ||\nabla_{\mathbf{s}_t} \log q_{\boldsymbol{\phi}}(\mathbf{s}_t | \mathbf{x}_t, \mathbf{a}_t, \mathbf{e}) - \nabla_{\mathbf{s}_t} \log p_{\boldsymbol{T}, \boldsymbol{\lambda}}(\mathbf{s}_t | \mathbf{a}_t, \mathbf{e}) ||^2 \right] \right]. \tag{19}$$

In practice, as described in Lu et al. (2022), we can jointly learn $(\boldsymbol{\theta}, \boldsymbol{\phi})$ by combining Eq. (18) and Eq. (19) in the following objective:

$$\mathcal{L}_{\text{phase1}}(\boldsymbol{\theta}, \boldsymbol{\phi}) = \mathcal{L}_{\text{phase1}}^{\text{ELBO}}(\boldsymbol{f}, \hat{\boldsymbol{T}}, \hat{\boldsymbol{\lambda}}, \boldsymbol{\phi}) - \mathcal{L}_{\text{phase1}}^{\text{SM}}(\hat{\boldsymbol{f}}, \boldsymbol{T}, \boldsymbol{\lambda}, \hat{\boldsymbol{\phi}}),$$

where $\hat{\boldsymbol{f}}, \hat{\boldsymbol{T}}, \hat{\boldsymbol{\lambda}}, \hat{\boldsymbol{\phi}}$ are copies of $\boldsymbol{f}, \boldsymbol{T}, \boldsymbol{\lambda}, \boldsymbol{\phi}$ that are treated as constants and whose gradient is not calculated during learning.

Importantly, under Assumptions 2b&3, we can follow the above steps to identify the latent variables $\mathbf{s}_t$ in NF-iVAE with similar theoretical guarantees on identifiability, by directly substituting $(\mathbf{x}, \mathbf{z}, \mathbf{y})$ with $(\mathbf{x}_t, \mathbf{s}_t, \mathbf{a}_t)$ in all the corresponding equations and theorems. All these are given in Appendix H.

### F.3.2 PHASE 2: DISCOVERING DIRECT CAUSES

After estimating $\mathbf{s}_t$ for $\mathbf{x}_t$, we follow Lu et al. (2022) and use a two-step approach to discover the direct causes $\mathbf{s}_t^p$ of the action $\mathbf{a}_t$. Roughly speaking, we first conduct the PC algorithm (Spirtes et al., 2000) to learn a Markov equivalence class of DAGs, which outputs the direct neighbours of $\mathbf{a}_t$, denoted by $\mathbf{NE}(\mathbf{a}_t)$. Then, all pairs in $\mathbf{NE}(\mathbf{a}_t)$ are tested with conditional independence testing (Zhang et al., 2012) to discover $\mathbf{z}_p$ by comparing their dependences after additionally conditioning on $\mathbf{a}_t$, cf. Lu et al. (2022) and Appendix C.1. It is worth noting that whether $\mathbf{c}_{t-1}$ is included or not, this approach works, because we concern only about the direct causes of $\mathbf{a}_t$.

### F.3.3 PHASE 3: LEARNING AN INVARIANT POLICY

After having obtained $\mathbf{s}_t^p$ for $\mathbf{a}_t$ across $\mathcal{E}_{tr}$, under Assumption 2c, the invariant predictor $w$ can be learned by solving the following optimization problem:

$$\min_w \sum_{e \in \mathcal{E}_{tr}} \mathbb{E}_{(\mathbf{s}_t^p)^e, \mathbf{a}_t} \left[ \ell(w((\mathbf{s}_t^p)^e), \mathbf{a}_t) \right], \tag{20}$$

where $\ell(\cdot)$ could be any loss. Since $p(\mathbf{a}_t | \mathbf{s}_p)$ is assumed to be invariant across $\mathcal{E}_{all}$, the learned $w$ is guaranteed to perform well across $\mathcal{E}_{all}$. See the theorems in Appendix H.

When in an unseen testing environment, $\mathbf{s}_t^p$ is required to be inferred from $\mathbf{x}_t$ before applying $w$ for prediction. This is done by leveraging the invariant $p_{\boldsymbol{f}}(\mathbf{x}_t | \mathbf{s}_t)$ (Assumption 2b) through solving the following optimization problem:

$$\Phi(\mathbf{x}_t) = \arg\max_{\mathbf{s}_t^p, \mathbf{s}_t^c} \log p_{\boldsymbol{f}}(\mathbf{x}_t | \mathbf{s}_t^p, \mathbf{s}_t^c) + \lambda_1 ||\mathbf{s}_t^p||_2^2 + \lambda_2 ||\mathbf{s}_t^c||_2^2, \tag{21}$$

where the parameters $\boldsymbol{f}$ learned from $\mathcal{E}_{tr}$ are fixed, and the hyperparameters $\lambda_1, \lambda_2 > 0$ control that the learned $\mathbf{s}_t^p$ and $\mathbf{s}_t^c$ have a reasonable scale, both of which are selected on training/validation data. Note that, here $\Phi$ can be viewed as a representation function that gives $\mathbf{s}_t^p$ from $\mathbf{x}_t$.

### F.4 MORE ON THE CONFOUNDER $\mathbf{C}_{t-1}$

Following the above, it is straightforward to obtain a corresponding generative model:

$$p_{\boldsymbol{\theta}}(\mathbf{x}_t, \mathbf{s}_t | \mathbf{a}_t, \mathbf{c}_{t-1}, \mathbf{e}) = p_{\boldsymbol{f}}(\mathbf{x}_t | \mathbf{s}_t) p_{\boldsymbol{T}, \boldsymbol{\lambda}}(\mathbf{s}_t | \mathbf{a}_t, \mathbf{c}_{t-1}, \mathbf{e}), \tag{22}$$

$$p_{\boldsymbol{f}}(\mathbf{x}_t | \mathbf{s}_t) = p_{\boldsymbol{\epsilon}}(\mathbf{x}_t - \boldsymbol{f}(\mathbf{s}_t)). \tag{23}$$

The corresponding evidence lower bound (ELBO) is

$$\mathcal{L}_{\text{phase1}}^{\text{ELBO}}(\boldsymbol{\theta}, \boldsymbol{\phi}) := \mathbb{E}_{\mathcal{D}_{\mathcal{E}_{tr}}} \left[ \mathbb{E}_{q_{\boldsymbol{\phi}}(\mathbf{s}_t | \mathbf{x}_t, \mathbf{a}_t, \mathbf{c}_{t-1}, \mathbf{e})} \left[ \log p_{\boldsymbol{f}}(\mathbf{x}_t | \mathbf{s}_t) + \log p_{\boldsymbol{T}, \boldsymbol{\lambda}}(\mathbf{s}_t | \mathbf{a}_t, \mathbf{c}_{t-1}, \mathbf{e}) - \log q_{\boldsymbol{\phi}}(\mathbf{s}_t | \mathbf{x}_t, \mathbf{a}_t, \mathbf{c}_{t-1}, \mathbf{e}) \right] \right], \tag{24}$$

where $q_{\boldsymbol{\phi}}(\mathbf{s}_t | \mathbf{x}_t, \mathbf{a}_t, \mathbf{c}_{t-1}, \mathbf{e})$ is an approximate conditional distribution for $\mathbf{s}_t$ given by a recognition network with parameters $\boldsymbol{\phi}$. To guarantee the identifiability result, the prior $p_{\boldsymbol{T}, \boldsymbol{\lambda}}(\mathbf{s}_t | \mathbf{a}_t, \mathbf{c}_{t-1}, \mathbf{e})$ is assumed to satisfy Assumption 3 (i.e. Eq. (15)). Considering the unknown normalization constant $\mathcal{Z}$ in this prior, we use score matching (Hyvärinen, 2005; Vincent, 2011) to learn $(\boldsymbol{T}, \boldsymbol{\lambda})$ by minimizing

$$\mathcal{L}_{\text{phase1}}^{\text{SM}}(\boldsymbol{T}, \boldsymbol{\lambda}) := \mathbb{E}_{\mathcal{D}_{\mathcal{E}_{tr}}} \left[ \mathbb{E}_{q_{\boldsymbol{\phi}}(\mathbf{s}_t | \mathbf{x}_t, \mathbf{a}_t, \mathbf{c}_{t-1}, \mathbf{e})} \left[ ||\nabla_{\mathbf{s}_t} \log q_{\boldsymbol{\phi}}(\mathbf{s}_t | \mathbf{x}_t, \mathbf{a}_t, \mathbf{c}_{t-1}, \mathbf{e}) - \nabla_{\mathbf{s}_t} \log p_{\boldsymbol{T}, \boldsymbol{\lambda}}(\mathbf{s}_t | \mathbf{a}_t, \mathbf{c}_{t-1}, \mathbf{e}) ||^2 \right] \right]. \tag{25}$$

In practice, as described in Lu et al. (2022), we can jointly learn $(\boldsymbol{\theta}, \boldsymbol{\phi})$ by combining Eq. (24) and Eq. (25) in the following objective:

$$\mathcal{L}_{\text{phase1}}(\boldsymbol{\theta}, \boldsymbol{\phi}) = \mathcal{L}_{\text{phase1}}^{\text{ELBO}}(\boldsymbol{f}, \hat{\boldsymbol{T}}, \hat{\boldsymbol{\lambda}}, \boldsymbol{\phi}) - \mathcal{L}_{\text{phase1}}^{\text{SM}}(\hat{\boldsymbol{f}}, \boldsymbol{T}, \boldsymbol{\lambda}, \hat{\boldsymbol{\phi}}),$$

where $\hat{\boldsymbol{f}}, \hat{\boldsymbol{T}}, \hat{\boldsymbol{\lambda}}, \hat{\boldsymbol{\phi}}$ are copies of $\boldsymbol{f}, \boldsymbol{T}, \boldsymbol{\lambda}, \boldsymbol{\phi}$ that are treated as constants and whose gradient is not calculated during learning.

## G    GENERALIZATION IN REINFORCEMENT LEARNING

### G.1    MORE EXPLANATIONS ON ASSUMPTION 4

The practicality of Assumption 4 can be detailedly explained in a similar way to that of Assumption 2. Importantly, it is worth noting here that we do not include that $r_{t+1} \perp\!\!\!\perp \mathbf{e} | \mathbf{s}_t^p$ (Fig. 4 in green, dashed line). In other words, $p(r_{t+1} | \mathbf{s}_t^p, \mathbf{a}_t)$ might change across $\mathcal{E}_{all}$ in various ways. Technically, in an environment $e \in \mathcal{E}_{all}$, the causal module $p^e(r_{t+1} | \mathbf{s}_t^p, \mathbf{a}_t) \doteq p(r_{t+1} | \mathbf{s}_t^p, \mathbf{a}_t, \mathbf{e} = e)$ can be represented by the following structural causal model (SCM) (Pearl, 2009):

$$r_{t+1}^e = f((\mathbf{s}_t^p)^e, \mathbf{a}_t, \epsilon^e; \boldsymbol{\theta}^e), \tag{26}$$

where $\epsilon$ is a disturbance term and has a non-zero variance (i.e., the model is not deterministic), and $\boldsymbol{\theta}$ denotes the effective parameters in the model/mechanism $f$. That they all, except $\mathbf{a}_t$, have the superscript $e$ explicitly indicates that all of them could be affected by $\mathbf{e}$ (i.e., they might vary across $\mathcal{E}_{all}$). Any change on these three terms on the RHS of Eq. (26) in the new environment $e^* \in \mathcal{E}_{all}$ will produce a different distribution $p^{e^*}(r_{t+1} | \mathbf{s}_t^p, \mathbf{a}_t)$, i.e., a different reward function $r_{t+1}^{e^*}$. The different ways the reward changes lead to the three types of generalization problems we study in the RL setting, which will be respectively discussed in the subsequent sections.

### G.2    IDENTIFYING LATENT VARIABLES IN THE RL SETTING

Under Assumption 4b, it is straightforward to obtain a corresponding generative model by directly substituting $(\mathbf{x}, \mathbf{z}, \mathbf{y})$ with $(\mathbf{x}_t, \mathbf{s}_t, r_{t+1})$ in Eqs. (4-5):

$$p_{\boldsymbol{\theta}}(\mathbf{x}_t, \mathbf{s}_t | r_{t+1}, \mathbf{e}) = p_{\boldsymbol{f}}(\mathbf{x}_t | \mathbf{s}_t) p_{\boldsymbol{T}, \boldsymbol{\lambda}}(\mathbf{s}_t | r_{t+1}, \mathbf{e}), \tag{27}$$

$$p_{\boldsymbol{f}}(\mathbf{x}_t | \mathbf{s}_t) = p_{\boldsymbol{\epsilon}}(\mathbf{x}_t - \boldsymbol{f}(\mathbf{s}_t)). \tag{28}$$

The corresponding evidence lower bound (ELBO) is

$$\mathcal{L}_{\text{phase1}}^{\text{ELBO}}(\boldsymbol{\theta}, \boldsymbol{\phi}) := \mathbb{E}_{\mathcal{D}_{\mathcal{E}_{tr}}} \left[ \mathbb{E}_{q_{\boldsymbol{\phi}}(\mathbf{s}_t | \mathbf{x}_t, r_{t+1}, \mathbf{e})} [\log p_{\boldsymbol{f}}(\mathbf{x}_t | \mathbf{s}_t) + \log p_{\boldsymbol{T}, \boldsymbol{\lambda}}(\mathbf{s}_t | r_{t+1}, \mathbf{e}) - \log q_{\boldsymbol{\phi}}(\mathbf{s}_t | \mathbf{x}_t, r_{t+1}, \mathbf{e})] \right], \tag{29}$$

where $q_{\boldsymbol{\phi}}(\mathbf{s}_t | \mathbf{x}_t, r_{t+1}, \mathbf{e})$ is an approximate conditional distribution for $\mathbf{s}_t$ given by a recognition network with parameters $\boldsymbol{\phi}$. To guarantee the identifiability result, the prior $p_{\boldsymbol{T}, \boldsymbol{\lambda}}(\mathbf{s}_t | r_{t+1}, \mathbf{e})$ is assumed to satisfy Assumption 3 (i.e. Eq. (15)). Considering the unknown normalization constant $\mathcal{Z}$ in this prior, we use score matching (Hyvärinen, 2005; Vincent, 2011) to learn $(\boldsymbol{T}, \boldsymbol{\lambda})$ by minimizing

$$\mathcal{L}_{\text{phase1}}^{\text{SM}}(\boldsymbol{T}, \boldsymbol{\lambda}) := \mathbb{E}_{\mathcal{D}_{\mathcal{E}_{tr}}} \left[ \mathbb{E}_{q_{\boldsymbol{\phi}}(\mathbf{s}_t | \mathbf{x}_t, r_{t+1}, \mathbf{e})} \left[ ||\nabla_{\mathbf{s}_t} \log q_{\boldsymbol{\phi}}(\mathbf{s}_t | \mathbf{x}_t, r_{t+1}, \mathbf{e}) - \nabla_{\mathbf{s}_t} \log p_{\boldsymbol{T}, \boldsymbol{\lambda}}(\mathbf{s}_t | r_{t+1}, \mathbf{e})||^2 \right] \right]. \tag{30}$$

In practice, as described in Lu et al. (2022), we can jointly learn $(\boldsymbol{\theta}, \boldsymbol{\phi})$ by combining Eq. (29) and Eq. (30) in the following objective:

$$\mathcal{L}_{\text{phase1}}(\boldsymbol{\theta}, \boldsymbol{\phi}) = \mathcal{L}_{\text{phase1}}^{\text{ELBO}}(\boldsymbol{f}, \hat{\boldsymbol{T}}, \hat{\boldsymbol{\lambda}}, \boldsymbol{\phi}) - \mathcal{L}_{\text{phase1}}^{\text{SM}}(\hat{\boldsymbol{f}}, \boldsymbol{T}, \boldsymbol{\lambda}, \hat{\boldsymbol{\phi}}),$$

where $\hat{\boldsymbol{f}}, \hat{\boldsymbol{T}}, \hat{\boldsymbol{\lambda}}, \hat{\boldsymbol{\phi}}$ are copies of $\boldsymbol{f}, \boldsymbol{T}, \boldsymbol{\lambda}, \boldsymbol{\phi}$ that are treated as constants and whose gradient is not calculated during learning.

### G.3    POLITY GENERALIZATION

Note that, unlike the IL setting where we learn the invariant policy in the supervised way, in the RL setting we follow Zhang et al. (2020b) and combine our MSR $\mathbf{s}_t^A$ with the soft actor-critic (SAC) algorithm (Haarnoja et al., 2018) to devise a practical RL method, termed *iCaRL-RL-P*, as shown in Algorithm 1 of the appendix. In principle, our MSR could be combined with any RL algorithm, including the model-free DQN (Mnih et al., 2015) or model-based PETS (Chua et al., 2018). When deploying the learned invariant policy in an unseen testing environment, we first infer $\mathbf{s}_t^A$ for $\mathbf{x}_t$ by leveraging the assumed invariant $p_{\boldsymbol{f}}(\mathbf{x}_t | \mathbf{s}_t)$ (Assumption 4b). This can be done by means of the

representation function $\Phi$ that is obtained by directly substituting $(\mathbf{s}_t^p, \mathbf{s}_t^c)$ with $(\mathbf{s}_t^A, \mathbf{s}_t^{-A})$ in Eq. (21), where $\mathbf{s}_t^{-A}$ are the remaining factors in $\mathbf{s}_t$ that are not the causal ancestors of the reward.

# H    THEOREMS AND PROOFS

Unless stated otherwise, it is straightforward that all the theorems presented in this section can be easily proved using the same technique in Lu et al. (2022). To avoid redundancy, we refer readers to Lu et al. (2022) for the proofs.

## H.1    GENERALIZATION IN IL

**Theorem 1.** *Assume that we observe data sampled from a generative model defined according to Eqs. (16, 17, 15), with parameters $\boldsymbol{\theta} := (\boldsymbol{f}, \boldsymbol{T}, \boldsymbol{\lambda})$, where $p_{\boldsymbol{T},\boldsymbol{\lambda}}(\mathbf{s}_t|\mathbf{a}_t, \mathbf{e})$ satisfies Assumption 3. Furthermore, assume the following holds: (i) The set $\{\mathbf{x}_t \in \mathcal{X}|\varphi_{\boldsymbol{\epsilon}}(\mathbf{x}_t) = 0\}$ has measure zero, where $\varphi_{\boldsymbol{\epsilon}}$ is the characteristic function of the density $p_{\boldsymbol{\epsilon}}$ defined in Eq. (17). (ii) Function $\boldsymbol{f}$ in Eq. (17) is injective, and has all second-order cross derivatives. (iii) The sufficient statistics in $\boldsymbol{T}_f$ are all twice differentiable. (iv) There exist $k + 1$ distinct points $(\mathbf{a}_t, \mathbf{e})^0, \ldots, (\mathbf{a}_t, \mathbf{e})^k$ such that the matrix $L = \left(\boldsymbol{\lambda}((\mathbf{a}_t, \mathbf{e})^1) - \boldsymbol{\lambda}((\mathbf{a}_t, \mathbf{e})^0), \ldots, \boldsymbol{\lambda}((\mathbf{a}_t, \mathbf{e})^k) - \boldsymbol{\lambda}((\mathbf{a}_t, \mathbf{e})^0)\right)$ of size $k \times k$ is invertible, where $k$ is the dimension of $\boldsymbol{T}$. Then the parameters $\boldsymbol{\theta}$ are identifiable up to a permutation and a* **"simple transformation"** *of the latent variables $\mathbf{s}$, defined as a componentwise nonlinearity making each recovered $\boldsymbol{T}_i(s_t^i)$ in $\boldsymbol{T}_f(\mathbf{s}_t)$ equal to the original up to a linear operation.*

**Theorem 2.** *Assume that the following holds: (i) The family of distributions $q_{\boldsymbol{\phi}}(\mathbf{s}_t|\mathbf{x}_t, \mathbf{a}_t, \mathbf{e})$ contains $p_{\boldsymbol{\theta}}(\mathbf{s}_t|\mathbf{x}_t, \mathbf{a}_t, \mathbf{e})$, and $q_{\boldsymbol{\phi}}(\mathbf{s}_t|\mathbf{x}_t, \mathbf{a}_t, \mathbf{e}) > 0$ everywhere. (ii) We maximize $\mathcal{L}_{phase1}(\boldsymbol{\theta}, \boldsymbol{\phi})$ with respect to both $\boldsymbol{\theta}$ and $\boldsymbol{\phi}$. Then in the limit of infinite data, we learn the true parameters $\boldsymbol{\theta}^*$ up to a permutation and simple transformation of the latent variables $\mathbf{s}_t$.*

**Theorem 3.** *Assume the hypotheses of Theorem 1 and Theorem 2 hold, then in the limit of infinite data, we identify the true latent variables $\mathbf{s}_t^*$ up to a permutation and simple transformation.*

**Proposition 4.** *Under Assumption 2 and the assumptions of Theorems 1 and 2, the imitation policy learned by iCaRL-IL across $\mathcal{E}_{tr}$ in the limit of infinite data has optimal OOD generalization across $\mathcal{E}_{all}$.*

### H.1.1    ON THE CONFOUNDER $\mathbf{c}_{t-1}$

**Theorem 4.** *Assume that we observe data sampled from a generative model defined according to Eqs. (16, 17, 15), with parameters $\boldsymbol{\theta} := (\boldsymbol{f}, \boldsymbol{T}, \boldsymbol{\lambda})$, where $p_{\boldsymbol{T},\boldsymbol{\lambda}}(\mathbf{s}_t|\mathbf{a}_t, \mathbf{c}_{t-1}, \mathbf{e})$ satisfies Assumption 3. Furthermore, assume the following holds: (i) The set $\{\mathbf{x}_t \in \mathcal{X}|\varphi_{\boldsymbol{\epsilon}}(\mathbf{x}_t) = 0\}$ has measure zero, where $\varphi_{\boldsymbol{\epsilon}}$ is the characteristic function of the density $p_{\boldsymbol{\epsilon}}$ defined in Eq. (17). (ii) Function $\boldsymbol{f}$ in Eq. (17) is injective, and has all second-order cross derivatives. (iii) The sufficient statistics in $\boldsymbol{T}_f$ are all twice differentiable. (iv) There exist $k + 1$ distinct points $(\mathbf{a}_t, \mathbf{c}_{t-1}, \mathbf{e})^0, \ldots, (\mathbf{a}_t, \mathbf{c}_{t-1}, \mathbf{e})^k$ such that the matrix $L = \left(\boldsymbol{\lambda}((\mathbf{a}_t, \mathbf{c}_{t-1}, \mathbf{e})^1) - \boldsymbol{\lambda}((\mathbf{a}_t, \mathbf{c}_{t-1}, \mathbf{e})^0), \ldots, \boldsymbol{\lambda}((\mathbf{a}_t, \mathbf{c}_{t-1}, \mathbf{e})^k) - \boldsymbol{\lambda}((\mathbf{a}_t, \mathbf{c}_{t-1}, \mathbf{e})^0)\right)$ of size $k \times k$ is invertible, where $k$ is the dimension of $\boldsymbol{T}$. Then the parameters $\boldsymbol{\theta}$ are identifiable up to a permutation and a* **"simple transformation"** *of the latent variables $\mathbf{s}$, defined as a componentwise nonlinearity making each recovered $\boldsymbol{T}_i(s_t^i)$ in $\boldsymbol{T}_f(\mathbf{s}_t)$ equal to the original up to a linear operation.*

**Theorem 5.** *Assume that the following holds: (i) The family of distributions $q_{\boldsymbol{\phi}}(\mathbf{s}_t|\mathbf{x}_t, \mathbf{a}_t, \mathbf{c}_{t-1}, \mathbf{e})$ contains $p_{\boldsymbol{\theta}}(\mathbf{s}_t|\mathbf{x}_t, \mathbf{a}_t, \mathbf{c}_{t-1}, \mathbf{e})$, and $q_{\boldsymbol{\phi}}(\mathbf{s}_t|\mathbf{x}_t, \mathbf{a}_t, \mathbf{c}_{t-1}, \mathbf{e}) > 0$ everywhere. (ii) We maximize $\mathcal{L}_{phase1}(\boldsymbol{\theta}, \boldsymbol{\phi})$ with respect to both $\boldsymbol{\theta}$ and $\boldsymbol{\phi}$. Then in the limit of infinite data, we learn the true parameters $\boldsymbol{\theta}^*$ up to a permutation and simple transformation of the latent variables $\mathbf{s}_t$.*

**Theorem 6.** *Assume the hypotheses of Theorem 4 and Theorem 5 hold, then in the limit of infinite data, we identify the true latent variables $\mathbf{s}_t^*$ up to a permutation and simple transformation.*

## H.2    GENERALIZATION IN RL

**Theorem 7.** *Assume that we observe data sampled from a generative model defined according to Eqs. (27, 28, 15), with parameters $\boldsymbol{\theta} := (\boldsymbol{f}, \boldsymbol{T}, \boldsymbol{\lambda})$, where $p_{\boldsymbol{T},\boldsymbol{\lambda}}(\mathbf{s}_t|r_{t+1}, \mathbf{e})$ satisfies Assumption 3. Furthermore, assume the following holds: (i) The set $\{\mathbf{x}_t \in \mathcal{X}|\varphi_{\boldsymbol{\epsilon}}(\mathbf{x}_t) = 0\}$ has measure zero, where $\varphi_{\boldsymbol{\epsilon}}$ is the characteristic function of the density $p_{\boldsymbol{\epsilon}}$ defined in Eq. (28). (ii) Function $\boldsymbol{f}$ in Eq. (28) is injective, and has all second-order cross derivatives. (iii) The sufficient statistics in $\boldsymbol{T}_f$ are all twice differentiable. (iv) There exist $k + 1$ distinct points $(r_{t+1}, \mathbf{e})^0, \ldots, (r_{t+1}, \mathbf{e})^k$ such that the matrix*

$L = \left(\boldsymbol{\lambda}((r_{t+1}, \mathbf{e})^1) - \boldsymbol{\lambda}((r_{t+1}, \mathbf{e})^0), \ldots, \boldsymbol{\lambda}((r_{t+1}, \mathbf{e})^k) - \boldsymbol{\lambda}((r_{t+1}, \mathbf{e})^0)\right)$ *of size $k \times k$ is invertible, where $k$ is the dimension of $\boldsymbol{T}$. Then the parameters $\boldsymbol{\theta}$ are identifiable up to a permutation and a* **"simple transformation"** *of the latent variables $\mathbf{s}$, defined as a componentwise nonlinearity making each recovered $\boldsymbol{T}_i(s_t^i)$ in $\boldsymbol{T}_f(\mathbf{s}_t)$ equal to the original up to a linear operation.*

**Theorem 8.** *Assume that the following holds: (i) The family of distributions $q_{\boldsymbol{\phi}}(\mathbf{s}_t|\mathbf{x}_t, r_{t+1}, \mathbf{e})$ contains $p_{\boldsymbol{\theta}}(\mathbf{s}_t|\mathbf{x}_t, r_{t+1}, \mathbf{e})$, and $q_{\boldsymbol{\phi}}(\mathbf{s}_t|\mathbf{x}_t, r_{t+1}, \mathbf{e}) > 0$ everywhere. (ii) We maximize $\mathcal{L}_{phase1}(\boldsymbol{\theta}, \boldsymbol{\phi})$ with respect to both $\boldsymbol{\theta}$ and $\boldsymbol{\phi}$. Then in the limit of infinite data, we learn the true parameters $\boldsymbol{\theta}^*$ up to a permutation and simple transformation of the latent variables $\mathbf{s}_t$.*

**Theorem 9.** *Assume the hypotheses of Theorem 1 and Theorem 2 hold, then in the limit of infinite data, we identify the true latent variables $\mathbf{s}_t^*$ up to a permutation and simple transformation.*

### H.2.1 POLICY GENERALIZATION IN RL

**Proposition 5.** *Under Assumptions 4,12&5 and the assumptions of Theorems 7&8, the policy learned across $\mathcal{E}_{tr}$ in the limit of infinite data has optimal OOD generalization across $\mathcal{E}_{all}$.*

*Proof.* This proof is straightforward. Under Assumptions 4&12 and the assumptions of Theorems 7&8, Theorems 9 guarantees that the latent variables $\mathbf{s}$ can be identified across $\mathcal{E}_{tr}$. Then, by leveraging the method in Phase 2, we can discover the minimum sufficient states $\mathbf{s}^A$. Since Assumption 5 indicates that all the environments share the same reward function and transition dynamics on top of $\mathbf{s}^A$, the learned policy across $\mathcal{E}_{tr}$ is guaranteed to generalize to $\mathcal{E}_{all}$. $\square$

### H.2.2 REPRESENTATION GENERALIZATION IN RL

**Proposition 6.** *Under Assumptions 4,12&6 and the assumptions of Theorems 7&8, the representation function $\Phi$ learned across $\mathcal{E}_{tr}$ in the limit of infinite data is able to generalize to $\mathcal{E}_{te}$.*

*Proof.* Under Assumptions 4&12 and the assumptions of Theorems 7&8, it is theoretically guaranteed that the latent variables $\mathbf{s}$ can be identified across $\mathcal{E}_{tr}$ and the representation function $\Phi$ can be also identified. Then, by leveraging the method in Phase 2, we can discover the minimum sufficient states $\mathbf{s}^A$. Also, under Assumption 4b, $\Phi$ is invariant across $\mathcal{E}_{all}$. Therefore, it is guaranteed that $\mathbf{s}^A$ can be inferred using $\Phi$ across $\mathcal{E}_{all}$. Under Assumption 6, we also know that all the environments share the same set of $\mathbf{s}^A$, which is the only information required for policy learning across $\mathcal{E}_{te}$. Hence, the learned representation function $\Phi$ is guaranteed to generalize to $\mathcal{E}_{te}$. $\square$

### H.2.3 DYNAMICS GENERALIZATION IN RL

**Proposition 7.** *Under Assumptions 4,12,7&8 and the assumptions of Theorems 7&8, the local dynamics model learned across $\mathcal{E}_{tr}$ in the limit of infinite data has optimal OOD generalization across $\mathcal{E}_{all}$.*

*Proof.* This proof is also straightforward. Under Assumptions 4&12 and the assumptions of Theorems 7&8, it is theoretically guaranteed that the latent variables $\mathbf{s}$ can be identified across $\mathcal{E}_{tr}$. Then, by leveraging the method in Phase 2, we can discover the direct causes $\mathbf{s}^{P_i}$ for each state $s^i$ under Assumption 7. Because Assumption 8 indicates that there exists some (local) dynamics model $p(s_{t+1}^i|\mathbf{s}_t^{P_i}, \mathbf{a}_t)$ that is invariant across $\mathcal{E}_{all}$. This shows that the learned local dynamics model has optimal generalization across $\mathcal{E}_{all}$. $\square$

## I EXPERIMENTAL RESULTS

### I.1 IDENTIFIABILITY ANALYSIS ON SYNTHETIC DATA

In this section, we empirically verify the identifiability of states in both IL and RL setting. For this, we conduct a series of experiments on synthetic data generated according to a family of MDPs, as shown in Fig. 3 of the appendix. Details of the ground truth data generating process are given in Appendix J.1. We train on two environments with different soft interventions on states. To measure identifiability, we compute the mean correlation coefficient (MCC) used in Khemakhem et al. (2020), which can be obtained by calculating the correlation coefficient between all pairs of true and recovered latent factors and then solving a linear sum assignment problem by assigning each

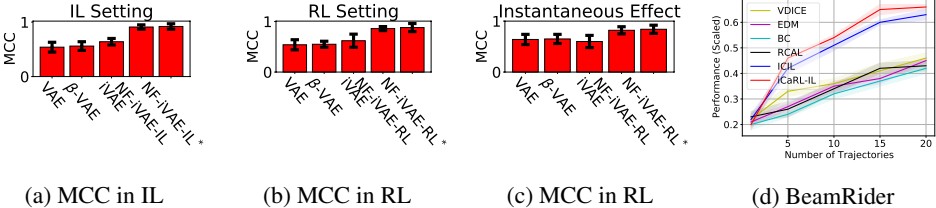

(a) MCC in IL      (b) MCC in RL      (c) MCC in RL      (d) BeamRider

Figure 7: (a-b) Mean correlation coefficient (MCC) scores for VAE, $\beta$-VAE, iVAE, NF-iVAE-I/RL, and NF-iVAE-I/RL$^*$ on synthetic data in IL and RL settings. Note that, NF-iVAE-I/RL$^*$ is a NF-iVAE-I/RL additionally conditioning on the previous observation-action pair $\mathbf{c}_{t-1}$. (c) Similar to (b), but with instantaneous effects between states on synthetic data. (d) Evaluation for policy generalization on OpenAI gym environments in the IL setting. $x$-axis indicates the number of trajectories with expert demonstrations from each training environment and $y$-axis is average return of the learned imitation policy on the test environment, scaled between 1 (expert performance) and 0 (random policy performance).

recovered latent factor to the true latent factor with which it best correlates. By definition, higher MCC scores indicate stronger identifiability. In the IL setting, Fig. 7a shows that the states are better identified by NF-iVAE-I/RL than by those commonly used VAE (Kingma & Welling, 2013) and $\beta$-VAE (Higgins et al., 2016). It is worth noting that when additionally conditioning on the previous observation-action pair $\mathbf{c}_{t-1}$, the result, denoted by NF-iVAE-IL$^*$, empirically confirms that it helps further identify the states. We can obtain the similar conclusion in the RL setting, as shown in Fig. 7b. Even when there exist instantaneous effects between state variables as mentioned in Section 2.1.2, we can see from Fig. 7c that NF-iVAE-RL works as well.

## I.2   GENERALIZATION IN IMITATION LEARNING

In the IL setting, we follow Bica et al. (2021) and evaluate generalization of our approach on the following two control tasks from OpenAI gym (Brockman et al., 2016): LunarLander (Brockman et al., 2016) and BeamRider (Bellemare et al., 2013) (Fig. 7d in the appendix). For each task, we also use pre-trained RL agents from RL Baselines Zoo (Raffin, 2018) and Stable OpenAI Baselines (Hill et al., 2018) to obtain expert policies, and then follow an approach similar to the one in Zhang et al. (2020a) to obtain datasets with demonstrations from the expert in two different environments. We follow the same setting of Bica et al. (2021) to add spurious correlations. We train on demonstrations from two training environments and test on an unseen testing environment. Further details can be found in Appendix J.2. Similar to Bica et al. (2021), we also compare our iCaRL-IL with 1) Behaviour Cloning (BC) (Pomerleau, 1991), 2) RCAL (Piot et al., 2014), 3) ValueDICE (VDICE) (Kostrikov et al., 2019), 4) Energy-based Distribution Matching (EDM) (Jarrett et al., 2020), 5) IRM (Arjovsky et al., 2019), and 6) ICIL (Bica et al., 2021). As shown in Fig. 2a, our iCaRL-IL consistently outperforms the benchmarks and can generalize better to the unseen testing environment.

## I.3   GENERALIZATION IN REINFORCEMENT LEARNING

### I.3.1   POLICY GENERALIZATION

In this section, we report experimental results on `cartpole_swingup` from the DeepMind Control (DMC) suite (Tassa et al., 2018) in two settings (i.e., *simple distractors* and *natural video distractors*, see Appendix J.3). we compare our iCaRL-RL-P against several baselines: 1) Stochastic Latent Actor-Critic (SLAC) (Lee et al., 2020), 2) DeepMDP (Gelada et al., 2019), 3) Model-Irrelevance State Abstractions (MISA) (Zhang et al., 2020a), 4) Deep Bisimulation for Control (DBC) (Zhang et al., 2020b), and 5) Invariant Policy Optimization (IPO) (Sonar et al., 2021). We first train policies on two `cartpole_swingup` environments with different *simple distractors* and then evaluate them on the average return obtained by deploying the learned policies on another `cartpole_swingup` environment with *natural video distractors*. Fig. 2b empirically verifies that the police learned by our method are able to generalize to the unseen testing environment.

### I.3.2   REPRESENTATION GENERALIZATION

We also evaluate the generalization performance of the learned representation function $\Phi$ on the DMC suite. Following Zhang et al. (2020b), this is done by training SAC with new reward functions on `walker_run` using the fixed representation $\Phi$ learned from two `walker_walk` environments with different *simple distractors*. Note that, here these new reward functions satisfy Assumption 6. As shown in Fig. 2c, our method, termed iCaRL-RL-R, learn a representation function which has better generalization power.

### I.3.3 DYNAMICS GENERALIZATION

We now test the generalization capacities of the learned dynamics on `cheetah_run` from the DMC suite. We first learn dynamics models on two `cheetah_run` environments with different *simple distractors* and then follow Zhang et al. (2020a) to evaluate them in terms of model errors obtained by deploying the learned dynamics models on another `cheetah_run` environment with *natural video distractors*. As shown in Fig. 2d, our method is more stable than the others in that the model error is consistently small.

## J MORE DETAILS ABOUT EXPERIMENTS

### J.1 SYNTHETIC DATA

The synthetic data are generated according to a family of MDPs, as shown in Fig. 3. Details of the ground truth data generating process are as follows:

$$\mathbf{e} \sim \mathcal{U}\{0.2, 2\}, \tag{31}$$

$$s_0^1 \sim \mathcal{N}(s_0^1|\mathbf{e}, 1), \tag{32}$$

$$s_0^2 \sim \mathcal{N}(s_0^2|\mathbf{e}, 1), \tag{33}$$

$$s_0^3 \sim \mathcal{N}(s_0^3|\mathbf{e}, 1), \tag{34}$$

$$\mathbf{a}_t \sim \mathcal{N}(\mathbf{a}_t|s_t^1 + s_t^2, 1), \tag{35}$$

$$s_{t+1}^1 \sim \mathcal{N}(s_{t+1}^1|\mathbf{a}_t + s_t^1 + E, 1), \tag{36}$$

$$s_{t+1}^2 \sim \mathcal{N}(s_{t+1}^2|\mathbf{a}_t + s_t^1 + s_t^2, 2), \tag{37}$$

$$s_{t+1}^3 \sim \mathcal{N}(s_{t+1}^3|s_t^2 + s_t^3 + 2E, 2), \tag{38}$$

$$r_{t+1} \sim \mathcal{N}(r_{t+1}|\mathbf{a}_t + 2s_t^2, 1), \tag{39}$$

$$\mathbf{x}_t = g(s_t^1, s_t^2, s_t^3), \tag{40}$$

where $\mathcal{U}\{\cdot\}$ denotes the discrete uniform distribution, $\mathcal{N}(\cdot)$ the Gaussian distribution, and $g(\cdot)$ is given by a neural network with 3-dimensional input and 10-dimensional output, whose parameters are randomly set in advance. We create a dataset consisting of samples from two training environments by drawing 2000 steps from each of the two environments $E = \{0.2, 2\}$.

Note that, in the experiments with instantaneous effects between states, we include an additional relationship between $s_t^2$ and $s_t^1$ by replacing Eq. (37) with

$$s_{t+1}^2 \sim \mathcal{N}(s_{t+1}^2|\mathbf{a}_t + s_t^1 + s_t^2 + 2s_{t+1}^1, 2). \tag{41}$$

In all the experiments on synthetic data, we set the number of the latent variables to $n = 3$. The architecture we used is as follows.

**NF-iVAE-I/RL $\lambda_f$-Nonlinear Prior**

- Input layer: Input batch *(batch size, input dimension)*
- Layer 1: Fully connected layer, output size = 6, activation = ReLU
- Output layer: Fully connected layer, output size = 3

**NF-iVAE-I/RL Encoder**

- Input layer: Input batch *(batch size, input dimension)*
- Layer 1: Fully connected layer, output size = 6, activation = ReLU
- Mean Output layer: Fully connected layer, output size = 3
- Log Variance Output layer: Fully connected layer, output size = 3

**NF-iVAE-I/RL Decoder**

- Input layer: Input batch *(batch size, input dimension)*
- Layer 1: Fully connected layer, output size = 6, activation = ReLU

- Mean Output layer: Fully connected layer, output size = output dimension
- Variance Output layer: $0.01 \times \mathbf{1}$, where $\mathbf{1}$ is a vector full of $1$ with the length of output dimension

**NF-iVAE-I/RL Predictor $w$**

- Input layer: Input batch *(batch size, input dimension)*
- Layer 1: Fully connected layer, output size = 6, activation = ReLU
- Output layer: Fully connected layer, output size = 1

## J.2 GENERALIZATION IN IMITATION LEARNING

Unless stated otherwise, we used the exact experimental setting and implementation details that are described in Bica et al. (2021) for the benchmarks. For a fair comparison, we use the same architecture for the encoder in our NF-iVAE-IL. We reverse the architecture of the encoder as a decoder. We set the number of the latent variables to $n = 50$. We do the hyperparameter search by exactly following the guides given in Bica et al. (2021). In addition, the architectures of the prior and the predictor are as below.

**NF-iVAE-IL $T_{NN}$-Prior**

- Input layer: Input batch *(batch size, input dimension)*
- Layer 1: Fully connected layer, output size = 50, activation = ReLU
- Output layer: Fully connected layer, output size = 45

**NF-iVAE-IL $\lambda_{NN}$-Prior**

- Input layer: Input batch *(batch size, input dimension)*
- Layer 1: Fully connected layer, output size = 50, activation = ReLU
- Output layer: Fully connected layer, output size = 45

**NF-iVAE-IL $\lambda_f$-Prior**

- Input layer: Input batch *(batch size, input dimension)*
- Layer 1: Fully connected layer, output size = 50, activation = ReLU
- Output layer: Fully connected layer, output size = 20

**Predictor $w$**

- Input layer: Input batch *(batch size, input dimension)*
- Layer 1: Fully connected layer, output size = 100, activation = ReLU
- Output layer: Fully connected layer, output size = 1

## J.3 GENERALIZATION IN REINFORCEMENT LEARNING

For convenience, here we restate the description of the two settings: *simple distractors* and *natural video distractors*. See more in Zhang et al. (2020b).

**Simple Distractors Setting.** We include simple background distractors, shown in Figure 3 of Zhang et al. (2020b) (middle row), with easy-to-predict motions. We use a fixed number of colored circles that obey the dynamics of an ideal gas (no attraction or repulsion between objects) with no collisions.

---

**Algorithm 1** iCaRL-RL-P

---

1:  **while** forever **do**
2:      **for** $e \in \mathcal{E}_{tr}$ **do**
3:          $\mathbf{a}_t \leftarrow \pi(\mathbf{x}_t^e)$
4:          $\mathbf{x}_{t+1}^e, r_{t+1} \leftarrow \mathtt{step}(\mathbf{x}_t^e, \mathbf{a}_t)$
5:          $\mathtt{store}(\mathbf{x}_t^e, \mathbf{a}_t, r_{t+1}, \mathbf{x}_{t+1}^e)$
6:      **end for**
7:      **for** $e \in \mathcal{E}_{tr}$ **do**
8:          Sample batch from replay buffer
9:          Estimate the representation function $\mathbf{s}_t^A = \Phi(\mathbf{x}_t)$ according to Section 3.2
10:     **end for**
11:     Train policy according to a modified SAC                          $\triangleright$ Algorithm 2
12: **end while**

---

**Algorithm 2** Train Policy (changes to SAC in red)

---

1:  Get value: $V = \min_{i=1,2} \hat{Q}_i(\Phi(\mathbf{x}_t)) - \alpha \log \pi(\mathbf{a}_t | \Phi(\mathbf{x}_t))$
2:  Train critics: $J(Q_i, \Phi) = (Q_i(\Phi(\mathbf{x}_t)) - r_{t+1} - \gamma V)^2$
3:  Train actor: $J(\pi) = \alpha \log p(\mathbf{a}_t | \Phi(\mathbf{x}_t)) - \min_{i=1,2} Q_i(\Phi(\mathbf{x}_t))$
4:  Train alpha: $J(\alpha) = -\alpha \log p(\mathbf{a}_t | \Phi(\mathbf{x}_t))$
5:  Update target critic: $\hat{Q}_i \leftarrow \tau_Q Q_i + (1 - \tau_Q)\hat{Q}_i$

---

**Natural Video Setting.**   We incorporate natural video from the Kinetics dataset (Kay et al., 2017) as background (Zhang et al., 2018), shown in Figure 3 of Zhang et al. (2020b) (bottom row).

Unless stated otherwise in the main text, we used the exact experimental setting and implementation details that are described in Bica et al. (2021) and Zhang et al. (2020a) for the benchmarks. For a fair comparison, we use the same architecture for the encoder in our NF-iVAE-RL. We reverse the architecture of the encoder as a decoder. We set the number of the latent variables to $n = 50$. We do the hyperparameter search by exactly following the guides given in Bica et al. (2021). In addition, the architectures of the prior and the predictor are as below.

**NF-iVAE-RL $T_{NN}$-Prior**

- Input layer: Input batch *(batch size, input dimension)*
- Layer 1: Fully connected layer, output size = 50, activation = ReLU
- Output layer: Fully connected layer, output size = 45

**NF-iVAE-RL $\lambda_{NN}$-Prior**

- Input layer: Input batch *(batch size, input dimension)*
- Layer 1: Fully connected layer, output size = 50, activation = ReLU
- Output layer: Fully connected layer, output size = 45

**NF-iVAE-RL $\lambda_f$-Prior**

- Input layer: Input batch *(batch size, input dimension)*
- Layer 1: Fully connected layer, output size = 50, activation = ReLU
- Output layer: Fully connected layer, output size = 20

**Predictor $w$**

- Input layer: Input batch *(batch size, input dimension)*
- Layer 1: Fully connected layer, output size = 100, activation = ReLU
- Output layer: Fully connected layer, output size = 1

