# OpenReview forum: "Invariant Causal Representation Learning for Generalization in Imitation and Reinforcement Learning"
_ICLR.cc/2022/Workshop/OSC — ICLR2022 OSC  Poster_

### Official Review · Reviewer_Ho2g · 2022-03-16

**Rating:** 2
**Confidence:** 1

**Review:**

Summary: the authors apply the Invariant Causal Representation Learning (iCaRL) framework to imitation and reinforcement learning.

Strengths:
- provides results on the conditions under which it is possible to expect generalization in representation learning, dynamics learning, and policy learning

Weaknesses:
- the method builds upon the non-linear ICA framework to identify latent variables. However, it is not clear in what sense these latent variables can be considered "causal" to the observed variables. As far as I know, from Elements of Causal Inference by Peters, Janzing, and Scholkopf, what makes a DAG "causal" is that it is the functions (i.e. mechanisms) that relate the variables are independent, and the paper was not clear how the non-linear ICA framework enforces the independence of mechanisms.

This paper was quite dense and had most of the results in the appendix. For example, the paper set up the imitation learning problem but did not present its proposed solution to this problem in the main text.

---

### Official Review · Reviewer_gYxy · 2022-03-17

**Rating:** 2
**Confidence:** 1

**Review:**

This paper proposes an approach to learn an invariant casual representation to improve generalization in both imitation and reinforcement learning. The paper is well positioned for this workshop, and provides extensive theoretical support for results

I unfortunately do not have much background on casual representation learning, and could not fully analyze the theoretical results. However -- I have a question for the authors. In practice, in many real world environments, there isn't a clear casual graph for a particular reward. For example, let's say the reward is making a cup a tea. What would be the appropriate casual representation to enable an agents to effectively make the cup of tea? How would you deal with partial observability in such a setting?

---

### Decision · Program_Chairs · 2022-03-19

**Decision:**

Accept (Poster)

**Comment:**

This paper is relevant to the workshop and provides strong theoretical as well as experimental support for the proposed contributions. Verifying the validity of the extensive theoretical contributions in the appendix of the paper is clearly out of scope for the workshop review process. For a workshop paper it is (as reviewer Ho2g pointed out) quite dense and we strongly encourage the authors to focus it instead on one or two particular contributions (instead of the current expansive list of 4 major contributions) when preparing the camera-ready version, to make it more accessible to a broader audience at the workshop.